# Deep Point Cloud Reconstruction

**Jaesung Choe**[1†]**, Byeongin Joung**[1†]**, Francois Rameau**[1]**, Jaesik Park**[2]**, In So Kweon**[1]

KAIST[1], POSTECH[2]

## Abstract

Point cloud obtained from 3D scanning is often sparse, noisy, and irregular. To cope with these issues, recent studies have been separately conducted to densify, denoise, and complete inaccurate point cloud. In this paper, we advocate that jointly solving these tasks leads to significant improvement for *point cloud reconstruction*. To this end, we propose a deep point cloud reconstruction network consisting of two stages: 1) a 3D sparse stacked-hourglass network as for the initial densification and denoising, 2) a refinement via transformers converting the discrete voxels into 3D points. In particular, we further improve the performance of transformer by a newly proposed module called *amplified positional encoding*. This module has been designed to differently amplify the magnitude of the positional encoding vectors based on the points' distances. Extensive experiments demonstrate that our network achieves state-of-the-art performance among the recent studies in the ScanNet, ICL-NUIM, and ShapeNetPart datasets. Moreover, we underline the ability of our network to generalize toward real-world and unmet scenes.

## 1 Introduction

3D scanning devices, such as LiDAR and RGB-D sensors, allows to quickly and accurately reconstruct a scene as a 3D point cloud. This compact 3D representation is commonly used to achieve various tasks in autonomous driving (Yang et al., 2018; Qi et al., 2018; Shi et al., 2020; Lang et al., 2019; Choe et al., 2021c;b; 2019b), robotics (He et al., 2021; 2020; Wang et al., 2019), or 3D mapping (Choy et al., 2020; Choe et al., 2021a;d). However, processing raw point cloud happens to be particularly challenging due to their sparsity, irregularity and sensitivity to noise (Xiang et al., 2019; Guo et al., 2020).

To address these issues, recent deep learning-based studies have proposed to improve the quality of point cloud. Specifically, prior works that are dedicated to point cloud refinement can be categorized into three distinct subfields: (1) point upsampling (Yu et al., 2018b), (2) point denoising (Rakotosaona et al., 2020), and (3) point completion (Yuan et al., 2018). Each of these tasks has its own benefits and inconveniences. For instance, the point

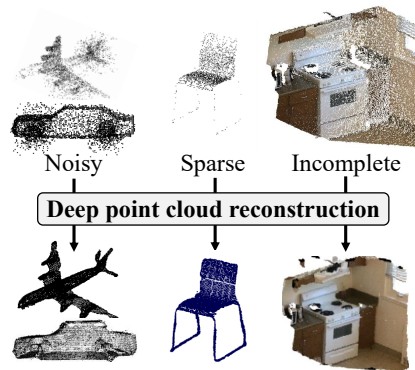

Figure 1: **Point cloud reconstruction.** We propose a novel neural architecture that jointly solves inherent shortcomings in raw point cloud, such as noise, sparsity, and incompleteness.

upsampling task usually handles noisy input points but do not explicitly deal with strong outliers. On the other hand, point denoising techniques have been designed to filter noisy data but cannot densify the point cloud. Finally, point completion methods mainly focus on object-level full 3D completion but are known for their weak generalization to unmet environments or unknown category of objects. Though there exist differences among the aforementioned tasks, their common goal is to improve proximity-to-surface in the 3D point representation.

Thus, we believe in the complementary of these three sub-tasks and hypothesize that a joint framework is greatly beneficial to deal with deteriorated and incomplete 3D point cloud. Accordingly, this paper challenges to resolve all these issues as a single task, called *point cloud reconstruction* as

---

† Both authors have equal contributions.

depicted in Fig. 1. To the best of our knowledge, this paper is the first attempt to jointly resolve the inherent shortcomings of point cloud obtained from 3D scanning devices: sparsity, noise, irregularity, and outliers. To this end, we propose a *deep point cloud reconstruction network* that consists in two stages: a voxel generation network and a voxel re-localization network.

In the first stage, the voxel generation network (Sec. 3.1) aims to densify voxels and remove outliers via sparse convolution layer (Choy et al., 2019). Rather than to use $k$-Nearest Neighbor that is sensitive to points' density (Mao et al., 2019), we utilize voxel hashing with sparse convolution layer to understand absolute-scale 3D structures for densification and denoising. Despite its success, voxelization inevitably leads to information loss due to the discretization process. To provide a fine-grained reconstruction, in the $2^{\text{nd}}$ stage, we propose a voxel re-localization network that converts discrete voxels into 3D points using transformer. Additionally, we increase the performance of transformer by the *amplified positional encoding*. In our analysis, spatial frequency plays an important role in the context of voxel-to-point conversion. Our new positional encoding strategy is useful to infer descriptive and detailed point cloud by simply changing the amplitude of the encoding vector. Our contributions can be summarized as follows:

- New problem formulation: point cloud reconstruction.
- Novel two-stage architecture for voxel-to-point refinement.
- A large number of experiments illustrating the generalization ability of our point cloud reconstruction to real 3D scans.

## 2 RELATED WORKS

Point cloud obtained from 3D scanning devices are known to contain various artifacts (Berger et al., 2017), such as noise, outliers, irregularity, and sparsity. Point cloud reconstruction aims at resolving these aforementioned issues in order to provide fine-grained 3D reconstructions. In this section, we introduce different approaches related to point cloud refinement.

**Point Cloud Denoising.** Without extra information, such as RGB images, point cloud denoising purely based on the input point distribution is a challenging task (Lee, 2000; Avron et al., 2010). Recent deep learning-based strategies (Rakotosaona et al., 2020; Roveri et al., 2018; Hermosilla et al., 2019; Pistilli et al., 2020; Luo & Hu, 2021) demonstrate promising results. However, despite the large improvement provided by these architectures, these methods do not have the ability to densify the reconstruction.

**Point Cloud Completion.** A family of solutions called point cloud completion have recently been proposed to alleviate the problem of incomplete reconstruction. One of the pioneer papers (Yuan et al., 2018) proposes to overcome the point incompleteness problem by generating points in both non-observed and visible areas. To improve the completion process, recent approaches (Huang et al., 2020; Zhang et al., 2020; Wen et al., 2021; Xiang et al., 2021) propose to include prior structural information of the objects. While these object-specific point completion techniques can lead to accurate results, they suffer from a lack of versatility since the object class is assumed to be known beforehand. Without such semantic level a priori, the entire completion in occluded surface (*e.g.*, behind objects) is a doubtful task. Thus, in this paper, we focus on a reliable and generic object-agnostic approach that is able to complete visible areas within an arbitrary point cloud and to cope with sparse, noisy, and unordered point cloud.

**Point Cloud Upsampling.** Given a sparse 3D reconstruction, the goal of point cloud upsampling is to densify the point cloud distribution. Conventional methods (Alexa et al., 2003; Huang et al., 2013) rely on point clustering techniques to re-sample points within the 3D space. Recent deep learning-based studies (Li et al., 2021; Yu et al., 2018a; Yifan et al., 2019; Yu et al., 2018b) and concurrent study Qiu et al. (2021) propose point densification networks under the supervision of dense ground truth points. Though some papers internally conduct point denoising (Yu et al., 2018a; Yifan et al., 2019), most of these works mainly assume that there are no strong outliers among the input points, such as flying-point cloud (You et al., 2019).

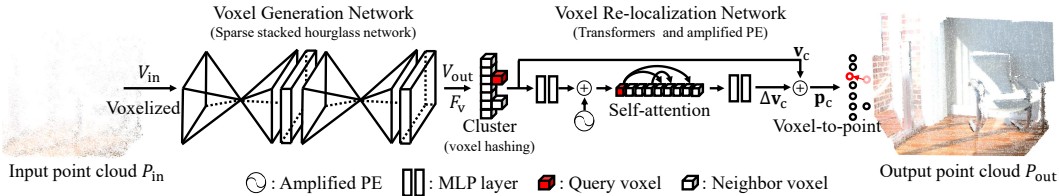

Figure 2: **Two-stage architecture for point cloud reconstruction.** In this figure, point cloud has been colorized for visualization purpose and we set the identical size of visualized points in $\mathcal{P}_{\text{in}}$ and $\mathcal{P}_{\text{out}}$ for fair comparison.

**Surface Reconstruction.** Point cloud meshing is a common solution for surface reconstruction (Hoppe et al., 1992; Curless & Levoy, 1996). For instance, this process can be achieved by converting point cloud to an implicit surface representation via signed distance function (SDF) and to apply Marching Cubes (Lorensen & Cline, 1987) or Dual Contouring (Ju et al., 2002) for meshing (*e.g.*, Poisson surface reconstruction (Kazhdan et al., 2006)). This strategies is known to be effective under favorable conditions but can hardly handle sparse and non-uniform points' distribution. Moreover, such technique tends to lead to over-smoothed and undetailed reconstruction (Berger et al., 2017). Similar limitations can be observed with recent deep learning-based SDF representation (Park et al., 2019). Many other solution have been developed for the problem of surface reconstruction, such as, Delaunay triangulation (Rakotosaona et al., 2021; Boissonnat, 1984; Kolluri et al., 2004), alpha shapes (Edelsbrunner & Mücke, 1994), or ball pivoting (Bernardini et al., 1999). Despite their compelling performances, these approaches require uniform, dense, and accurate inputs since they usually preserve the original distribution of a given point cloud. This assumption still holds for the recent studies (Sharp & Ovsjanikov, 2020; Rakotosaona et al., 2021).

Compared to surface reconstruction, point cloud reconstruction can be viewed as a prior-process to overcome natural artifacts: sparsity, noise, irregularity. Furthermore, in contrast to previous studies, our method are object-agnostic and generally applicable toward the real-world 3D point cloud. To this end, we introduce our deep point cloud reconstruction network that consists of the voxel generation network and the voxel re-localization network.

## 3 DEEP POINT CLOUD RECONSTRUCTION

Given a noisy and sparse point cloud $\mathcal{P}_{\text{in}}$, point cloud reconstruction aims to generate a dense and accurate set of points $\mathcal{P}_{\text{out}}$. For this purpose, we propose a deep point cloud reconstruction network composed of two stages: voxel generation network (Sec. 3.1) and voxel re-localization network (Sec. 3.2). The first network has been designed to perform a densification and outliers removal in the sparse voxel space. The second network complements and improves this first stage by denoising the data further and by converting voxels back to 3D points. The overall architecture is illustrated in Fig. 2.

### 3.1 VOXEL GENERATION NETWORK

Most previous studies (Li et al., 2021; Yifan et al., 2019; Yu et al., 2018a; Luo & Hu, 2021) attempts to resolve these problems by directly processing a raw point cloud. However, it is known to be a challenging problem due to the unordered nature of the 3D point cloud. To tackle this issue, we propose to convert the 3D point cloud $\mathcal{P}_{\text{in}}$ into a sparse voxel representation $\mathcal{V}_{\text{in}}$ through our voxel generation network. From this data, our network generates a refined and densified volume $\mathcal{V}_{\text{out}}$. A noticeable advantage of sparse voxel representation is its ability to preserve the neighbor connectivity of point cloud in a uniform and regular 3D grids (Guo et al., 2020) via voxel hashing (Choy et al., 2019; Graham & van der Maaten, 2017). Voxel hashing keeps tracking of the geometric positions of sparse tensors so that we can find the voxels' neighbors to apply sparse convolution. Moreover, sparse convolution layer can cover the consistent and absolute-scale scope of the local region and hierarchically capture its large receptive fields. Such property is then particularly beneficial to deal with an inaccurate and unordered input point cloud $\mathcal{P}_{\text{in}}$.

Figure 3: **Voxel generation network.** We design sparse stacked hourglass network that hierarchically and sequentially generates and prunes voxels in a coarse-to-fine manner.

In order to denoise and densify $\mathcal{V}_{in}$, our voxel generation network consists of a sparse stacked hourglass network as shown in Fig. 3. Our network architecture is close to the previous studies (Newell et al., 2016; Chang & Chen, 2018; Im et al., 2019; Choe et al., 2019a) in that it is composed of several hourglass networks for cascaded refinements of images or dense volumes. In the context of this paper, scanning devices can only reconstruct the surface of the structure, thus, the resulting voxelized volume is inherently sparse. Therefore, to process this 3D volume efficiently, our voxel generation network takes advantage of the sparse convolution network (Choy et al., 2019).

It should be noted that the specificity of sparse convolution operation is particularly desirable for the tasks at hand. First, generative sparse transposed convolution layer (Gwak et al., 2020) has the ability to create new voxels given a pre-defined upsampling rate $r$, which can be seen as a point upsampling methodology (Yu et al., 2018b). Secondly, voxel pruning is widely used for fine-grained volume rendering (Liu et al., 2020) and efficient computation (Choy et al., 2019). In our context, voxel pruning is closely related to outlier removal. Last, the joint use of generation and pruning enables our network to create an arbitrary number of points.

As illustrated in Fig. 3, we first compute local voxel features using sparse convolutional blocks in an encoder part. The aggregated voxel features undergo the voxel generation layers and pruning layers in a decoder part. We use two hourglass networks to densify and prune voxels. To train the voxel generation network, we adopt a Binary Cross-Entropy loss (BCE) to classify the status of estimated voxels into an occupied/empty state as:

$$\mathcal{L}_{BCE}(\mathcal{V}_{out}) = \frac{1}{\mathcal{N}_v} \sum_{\mathbf{v} \in \mathcal{V}_{out}} y_{\mathbf{v}} \cdot \log(\hat{y_{\mathbf{v}}}) + (1 - y_{\mathbf{v}}) \cdot \log(1 - \hat{y_{\mathbf{v}}}), \quad (1)$$

where $\hat{y_{\mathbf{v}}}$ is an estimated class of a generated voxel $\mathbf{v}$ from the network, and $y_{\mathbf{v}}$ is the ground truth class at the generated voxel $\mathbf{v}$. $\mathcal{N}_v$ is the total number of voxels that we used to compute the BCE loss. If $\mathbf{v}$ is on the 3D surface, $y_{\mathbf{v}}$ becomes true class (*i.e.*, $y_{\mathbf{v}}{=}1$). Furthermore, following the original loss design (Newell et al., 2016), we compute the BCE Loss $\mathcal{L}_{BCE}$ for both the final voxel prediction $\mathcal{V}_{out}$ and the intermediate results $\mathcal{V}_{mid}$. In total, we compute the voxel loss $\mathcal{L}_{vox}$ as:

$$\mathcal{L}_{vox} = \mathcal{L}_{BCE}(\mathcal{V}_{out}) + 0.5 \cdot \frac{1}{\mathcal{N}_v} \sum_{\mathcal{V}_i \in \mathcal{V}_{mid}} \mathcal{L}_{BCE}(\mathcal{V}_i), \quad (2)$$

where $\mathcal{N}_v$ is the total number of voxels in the intermediate inference volume $\mathcal{V}_{mid}$ that we used to calculate the BCE loss. Along with the predicted voxels $\mathcal{V}_{out}$, we extract voxel features $\mathcal{F}_{\mathbf{v}}$ using sparse interpolation as illustrated in Fig. 3.

## 3.2 VOXEL RE-LOCALIZATION NETWORK

Despite the denoising and densification offered by the $1^{st}$ stage network, voxelization inevitably leads to information loss. Thus, in this section, we propose a voxel re-localization network that converts the discrete voxels $\mathcal{V}_{out}{\in}\mathbb{R}^{N \times 3}$ into 3D points $\mathcal{P}_{out}{\in}\mathbb{R}^{N \times 3}$. This conversion requires an understanding of local geometry to describe the 3D surface as a group of point sets. Inspired by (Zhao et al., 2021; Choe et al., 2021d), we utilize self-attention as a point set operator.

Let us describe the detailed process in the voxel re-localization network, which is illustrated in Fig. 4. Given output voxels $\mathcal{V}_{out}$, we collect the $K(=8)$ closest voxels to each voxel $\mathbf{v}_i{\in}\mathbb{R}^{1 \times 3}$ using the hash table that we used in the $1^{st}$ stage network. Then, we obtain a voxel set $\mathcal{V}_i{=}\{\mathbf{v}_k\}_{k=1}^K$ that

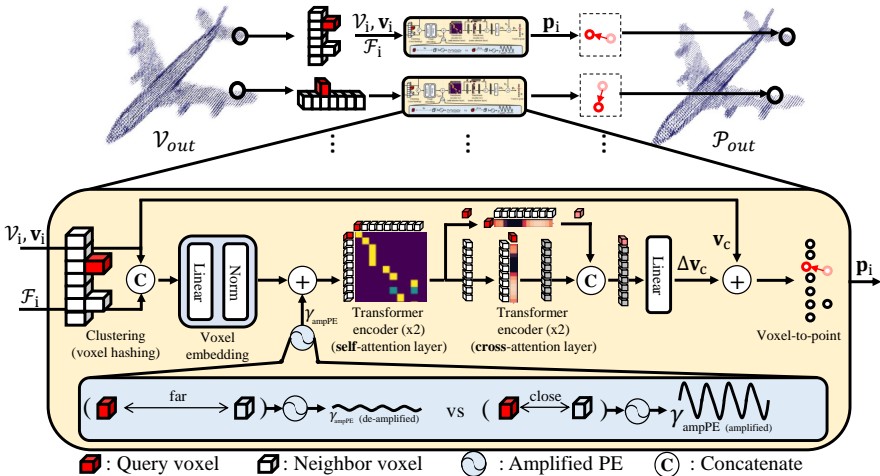

Figure 4: **Voxel re-localization network.** The $2^{nd}$ stage network involves transformers for voxel-to-point refinement. In particular, transformers are applied into self/cross-attention layers with our amplified positional encoding to compute the relation between a query voxel and its neighbor voxels.

is referenced to the query voxel $\mathbf{v}_i$. Similarly, we cluster a set of voxel features $\mathcal{F}_i = \{\mathbf{f}_k\}_{k=1}^{K}$ that correspond to $\mathcal{P}_i$. As in Fig. 4, the 2nd stage network regresses the location of a query point $\mathbf{p}_i$ (red voxel) using the geometric relation between the query voxel $\mathbf{v}_i$ and its neighbor $\mathcal{V}_i$ (white voxels) through self/cross attention layers.

While this formulation is close to PointTransformer (Zhao et al., 2021), our voxel re-localization network has three distinctive differences. First, PointTransformer utilizes $k$-Nearest Neighbors to group 3D points but we re-use the voxel hashing that covers absolute-scale local geometry. Second, our network consists of both self-attention and cross-attention. In our analysis, it is better to use two different attention layer than to solely adopt self-attention. Third, we design our own positional encoding system that controls the necessity of high-frequency signal by changing the amplitude of the encoding vector, called *amplified positional encoding*.

Originally, positional encoding has been introduced to help transformer to understand the orders of tokens (Vaswani et al., 2017). When applied to other domains of applications, positional encoding is often used to preserve high-frequency signals which tend to be disregarded by neural networks (Rahaman et al., 2019; Ramasinghe & Lucey, 2021). For example, positional encoding is proven to be essential to synthesize sharp images from an implicit 3D neural representation (NeRF (Mildenhall et al., 2020)). For our purpose, we need to consider spatial frequency to coherently re-locate a query voxel $\mathbf{v}_i$ by using its neighbor set $\mathcal{V}_i$. Our intuition is simple. Since positional encodings are mainly effective for the high-frequency embedding. As in Fig. 4, we increase the amplitude of positional encoding when we need to move a query voxel by a small amount of distance (*i.e.*, high-frequency in 3D spatial domain). If a query voxel is far away from its neighbor, we decrease the amplitude of the positional encoding. Based on the relation between high frequency signals and our voxel re-location scheme, we define our amplified positional encoding $\gamma_{\text{ampPE}}$ as:

$$\gamma_{\text{ampPE}}(\mathbf{d}, pos) = e^{-\|\mathbf{d}\|_1^1} \cdot \gamma_{\text{PE}}(pos), \quad (3)$$

$$\gamma_{\text{PE}}(pos) = [..., \sin(pos/10000^{i/C}), \cos(pos/10000^{i/C})], \quad (4)$$

where $\gamma_{\text{ampPE}}$ is an amplified positional encoding vectors and $\|\cdot\|_1^1$ is the L1-norm. Following the conventions (Vaswani et al., 2017), $\gamma_{\text{PE}}$ is 1D positional encoding vectors, $pos$ represents position, $i$ means dimension, and $C$ is channel dimensions of voxel embedding. After we encode $\gamma_{\text{ampPE}}$ into the voxel embedding, we aggregate this embedding through self- and cross-attention layers to capture local structures around the query voxel. Finally, the regressed offset $\Delta \mathbf{v}$ is added to the original location of query voxel $\mathbf{v}_c$ to update its location. To train the second stage network, we calculate a chamfer distance loss $\mathcal{L}_{\text{dist}}$ as:

$$\mathcal{L}_{\text{dist}} = \frac{1}{\mathcal{N}_{\mathcal{P}_{\text{pred}}}} \sum_{\tilde{\mathbf{p}} \in \mathcal{P}_{\text{pred}}} \left( \min_{\mathbf{p} \in \mathcal{P}_{\text{gt}}} \|\tilde{\mathbf{p}} - \mathbf{p}\|_2^2 \right) + \frac{1}{\mathcal{N}_{\mathcal{P}_{\text{gt}}}} \sum_{\mathbf{p} \in \mathcal{P}_{\text{gt}}} \left( \min_{\tilde{\mathbf{p}} \in \mathcal{P}_{\text{pred}}} \|\mathbf{p} - \tilde{\mathbf{p}}\|_2^2 \right). \quad (5)$$

where $\mathcal{P}_{\text{pred}}$ is the predicted point cloud, $\mathcal{P}_{\text{gt}}$ indicates ground truth point cloud, $\tilde{\mathbf{p}}$ represents a predicted point, and $\mathbf{p}$ means a ground truth point. $\mathcal{N}_{\mathcal{P}_{\text{pred}}}$ and $\mathcal{N}_{\mathcal{P}_{\text{gt}}}$ are the number of prediction and ground truth point cloud, respectively. Through our two-stage pipelines, our whole network performs point cloud reconstruction. By jointly generating, removing, and refining points, our network can robustly recover accurate point cloud and increase the level of proximity-to-surface.

## 4 EXPERIMENT

In this section, we describe the implementation details of our network and we present a large series of assessments underlying the effectiveness of our approach. Our point reconstruction network has been solely trained on the ShapeNet-Part (Yi et al., 2016) dataset but tested on other real and synthetic datasets such as ScanNet (Dai et al., 2017) and ICL-NUIM (Handa et al., 2014). These experiments highlight the ability of our solution to generalize well to other scenarios involving various levels of noise and different densities.

### 4.1 IMPLEMENTATION

We first train our voxel generation network for 10 epochs using the Adam optimizer (Kingma & Ba, 2014) with initial learning of $1e$-3 and a batch size of 4. We decrease the learning rate by half for every 2 epochs. After that, we freeze the voxel generation network and initiate the training of the voxel re-localization using the inferred voxels $\mathcal{V}_{\text{out}}$ from the voxel generation network. The point relocation network is trained using identical hyper-parameters used for training the first stage network. Further details regarding the training rules are provided in Sec. 4.3.

### 4.2 BASELINE APPROACHES

We conduct a thorough comparison against previous state-of-the-art techniques: **PU** for point cloud upsampling (Li et al., 2021), **PD** for point cloud denoising (Luo & Hu, 2021), and **PC** for point cloud completion (Xiang et al., 2021). These three papers are currently positioned at the highest rank for each task and we utilize their official implementation for training and evaluation. all methods are trained with the same data augmentation as competing approaches and its performances have been measured on various datasets: ShapeNet-part (Yi et al., 2016), ScanNet (Dai et al., 2017), and ICL-NUIM (Handa et al., 2014) datasets. In particular, we analyze the efficacy of ours compared to the combination of the existing studies. For instance in Table 1, $PC(r{=}4){\rightarrow}PU(r{=}4){\rightarrow}PD$ represents consecutive operation of point completion, and point upsampling and point denoising under upsampling ratio $r{=}4$. Note that we also take into account the point completion task since this task also considers point densification on visible surface.

### 4.3 DATASET

**ShapeNet dataset.** We utilize 13 different categories of objects categories from the ShapeNet-Part dataset (Chang et al., 2015; Yi et al., 2016). We extract dense and uniform 10K ground truth point cloud per object by using the Poisson Disk Sampling algorithm (Yuksel, 2015) provided by Open3D (Zhou et al., 2018). Among these ground truth 3D points, we randomly sample 2048 points as an input $\mathcal{P}_{\text{in}}$ for the training networks. For the sake of fairness, we adjust the unit voxel length as $l_{\text{vox}}{=}0.0200$ for training the networks. This is to produce a comparable number of points than previous studies. We follow a data augmentation scheme proposed by the previous point upsampling study (Li et al., 2021), such as random noise addition and random re-scaling. Additionally, we include random outliers that constitute less that 5 percent of input points. To train and validate the networks, we carefully follow the official train/val/test split provided by (Yi et al., 2016).

**ScanNet / ICL-NUIM dataset.** These datasets (Dai et al., 2017; Handa et al., 2014) are indoor datasets that provide RGB-D sequences and corresponding camera poses. We utilize these datasets to evaluate the generalization capability of ours and previous methods. Similar to the image-based reconstruction study (Yao et al., 2018), we firstly re-use the network parameters pre-trained in the ShapeNet dataset, then evaluate all methods in unmet datasets without fine-tuning networks. Further details are included in the supplementary material.

| | | | ShapeNet-Part dataset | | | | | | | | |
|---|---|---|---|---|---|---|---|---|---|---|---|
| | | | $k$-Chamf. ($\downarrow$) | | | Percentage ($\uparrow$) ($< 0.5$) | | | Percentage ($\uparrow$) ($< 1.0$) | | |
| Methods | $\mathcal{P}_{\text{in}}$ | $\mathcal{P}_{\text{out}}$ | $k{=}1$ | $k{=}2$ | $k{=}4$ | Acc. | Comp. | $f$-score | Acc. | Comp. | $f$-score |
| PC ($r{=}4$) | 2048 | 8192 | 1.14 | 1.29 | 1.51 | 72.28 | 42.71 | 51.51 | 92.41 | 88.48 | 90.16 |
| PU ($r{=}4$) | 2048 | 8192 | 1.56 | 1.69 | 1.90 | 60.71 | 26.91 | 36.04 | 86.67 | 68.19 | 75.60 |
| PointRecon ($l_{\text{vox}}{=}0.0200$) | 2048 | 8732 | 1.19 | 1.33 | 1.52 | 81.02 | 40.41 | 53.48 | 96.95 | 81.23 | 88.08 |
| PC ($r{=}4$) $\to$ PU ($r{=}4$) $\to$ PD | 2048 | 32987 | 1.14 | 1.48 | 1.60 | 56.55 | 49.72 | 52.43 | 87.13 | 82.73 | 84.67 |
| PU ($r{=}4$) $\to$ PC ($r{=}4$) $\to$ PD | 2048 | 32723 | 1.51 | 1.59 | 1.71 | 56.63 | 59.90 | 57.63 | 83.13 | 89.47 | 85.97 |
| PointRecon ($l_{\text{vox}}{=}0.0150$) | 2048 | 15863 | 0.90 | 0.94 | 1.12 | 85.77 | 69.01 | 78.14 | 97.11 | 94.98 | 96.16 |

| | | | ScanNet dataset | | | | | | | | |
|---|---|---|---|---|---|---|---|---|---|---|---|
| | | | $k$-Chamf. ($\downarrow$) | | | Percentage ($\uparrow$) ($< 0.5$) | | | Percentage ($\uparrow$) ($< 1.0$) | | |
| Methods | $\mathcal{P}_{\text{in}}$ | $\mathcal{P}_{\text{out}}$ | $k{=}1$ | $k{=}2$ | $k{=}4$ | Acc. | Comp. | $f$-score | Acc. | Comp. | $f$-score |
| PC ($r{=}16$) | 4098 | 65536 | 3.79 | 4.08 | 4.48 | 16.91 | 11.96 | 13.12 | 40.45 | 53.70 | 45.45 |
| PC ($r{=}4$) $\to$ PU ($r{=}4$) $\to$ PD | 4096 | 64427 | 3.69 | 3.91 | 4.23 | 23.05 | 12.59 | 15.31 | 49.75 | 42.24 | 44.84 |
| PU ($r{=}16$) | 4096 | 65536 | 3.68 | 3.94 | 4.32 | 16.81 | 13.48 | 13.90 | 37.89 | 44.60 | 40.04 |
| PU ($r{=}4$) $\to$ PC ($r{=}4$) $\to$ PD | 4096 | 64578 | 2.66 | 2.86 | 3.14 | 22.89 | 23.30 | 21.86 | 49.64 | 64.11 | 55.40 |
| PU ($r{=}4$) $\to$ PD $\to$ PC ($r{=}4$) | 4096 | 63118 | 3.75 | 3.99 | 4.33 | 15.99 | 15.43 | 14.50 | 36.60 | 54.70 | 43.10 |
| PointRecon ($l_{\text{vox}}{=}0.0075$) | 4096 | 55188 | 1.91 | 2.04 | 2.24 | 57.88 | 36.83 | 44.72 | 88.06 | 68.34 | 76.72 |

| | | | ICL-NUIM dataset | | | | | | | | |
|---|---|---|---|---|---|---|---|---|---|---|---|
| | | | $k$-Chamf. ($\downarrow$) | | | Percentage ($\uparrow$) ($< 0.5$) | | | Percentage ($\uparrow$) ($< 1.0$) | | |
| Methods | $\mathcal{P}_{\text{in}}$ | $\mathcal{P}_{\text{out}}$ | $k{=}1$ | $k{=}2$ | $k{=}4$ | Acc. | Comp. | $f$-score | Acc. | Comp. | $f$-score |
| PC ($r{=}16$) | 4098 | 65536 | 3.41 | 3.63 | 3.95 | 18.69 | 17.53 | 16.65 | 38.26 | 63.02 | 47.20 |
| PC ($r{=}4$) $\to$ PU ($r{=}4$) $\to$ PD | 4096 | 64451 | 3.61 | 3.81 | 4.11 | 21.62 | 11.41 | 14.05 | 45.34 | 42.57 | 43.32 |
| PU ($r{=}16$) | 4096 | 65536 | 3.83 | 4.03 | 4.33 | 17.78 | 18.19 | 16.83 | 35.45 | 49.93 | 41.03 |
| PU ($r{=}4$) $\to$ PC ($r{=}4$) $\to$ PD | 4096 | 64872 | 3.61 | 3.94 | 4.42 | 27.55 | 8.64 | 12.42 | 54.42 | 33.58 | 40.60 |
| PU ($r{=}4$) $\to$ PD $\to$ PC ($r{=}4$) | 4096 | 63521 | 4.00 | 4.21 | 4.53 | 10.09 | 11.64 | 9.51 | 22.70 | 48.52 | 30.25 |
| PointRecon ($l_{\text{vox}}{=}0.0075$) | 4096 | 41322 | 1.87 | 1.99 | 2.18 | 67.78 | 44.97 | 53.36 | 89.09 | 70.52 | 78.141 |
| PointRecon ($l_{\text{vox}}{=}0.0065$) | 4096 | 56902 | 2.14 | 2.26 | 2.45 | 63.40 | 42.85 | 80.26 | 86.84 | 67.78 | 75.41 |
| PointRecon ($l_{\text{vox}}{=}0.0050$) | 4096 | 99321 | 2.78 | 2.92 | 3.10 | 54.11 | 38.09 | 43.55 | 81.76 | 61.88 | 69.45 |

Table 1: **Quantitative results of point cloud reconstruction.** Note that PC, PU, PD represent point completion (Xiang et al., 2021), point upsampling (Li et al., 2021), and point denoising (Luo & Hu, 2021), respectively. Please refer to the appendix for more results.

## 4.4 COMPARISON

For metric computation, we adopt Chamfer distance and percentage metrics (*e.g.*, Accuracy, Completeness, and $f$-score) (Yao et al., 2018; Jensen et al., 2014), as shown in Table 1. In particular, we extend to measure the $k$-Nearest Chamfer distance (*i.e.*, $k$-Chamf.[1]) that efficiently computes point-wise distance than Earth Mover's distance. Based on these criteria, we calculate the metrics of the whole methods.

As shown in Table 1, our network achieves state-of-the-art performance compared to the previous studies (Li et al., 2021; Luo & Hu, 2021; Xiang et al., 2021). Moreover, the performance gap increases when we conduct generalization test in ScanNet dataset and ICL-NUIM dataset. There are two dominant reasons behind the better generalization and transferability. First, each sparse convolution layer has consistent and absolute-scale receptive fields in 3D space. This local voxel neighborhood appears to be more robust than raw point cloud processing via $k$-Nearest Neighbor as adopted in the three baselines (Xiang et al., 2021; Luo & Hu, 2021; Li et al., 2021). This is also one of the strength of using voxelization as an initial processing of point cloud reconstruction, which is conducted by our $1^{\text{st}}$ network. Second, our network does not have task-specific header networks or losses for completion/upsampling/denoising tasks. Instead, Our network purely focuses on the inherent problems resided in raw point cloud. Such joint approach trains our network to improve the overall proximity-to-surface, not the task-specific quality.

Regardless of the combined use of the baselines, our method systematically outperforms them. When we sequentially apply point densification and point denoising, the performances are not constantly improved in Table 1. This observation is supported by the qualitative results in Fig. 5. While point denoising can effectively remove the remaining outliers that are generated from (Xiang et al., 2021) or (Li et al., 2021), it has chance to remove the wrong points sets. For instance in Fig. 4, the point denoising method erases part of the airplane's wing. In contrast, our voxel generation network conducts both voxel pruning and generation in each hourglass network so that it can effectively circumvent such case.

---

[1]Precise equations of the metric computation are described in the appendix.

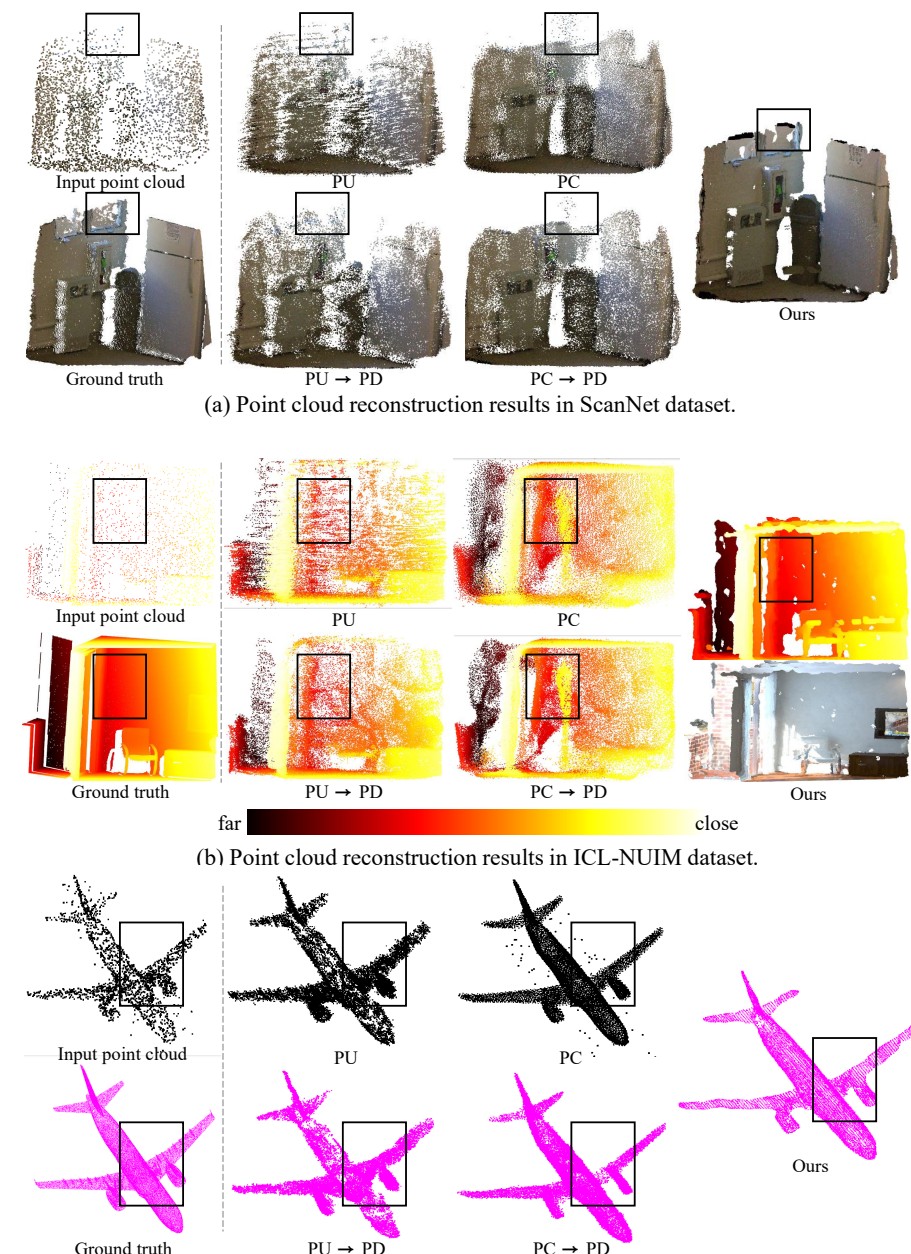

(a) Point cloud reconstruction results in ScanNet dataset.

(b) Point cloud reconstruction results in ICL-NUIM dataset.

(c) Point cloud reconstruction results in ShapeNetPart dataset.

Figure 5: **Point cloud reconstruction results using two baselines.** Note that point cloud has been colorized for a visualization purpose.

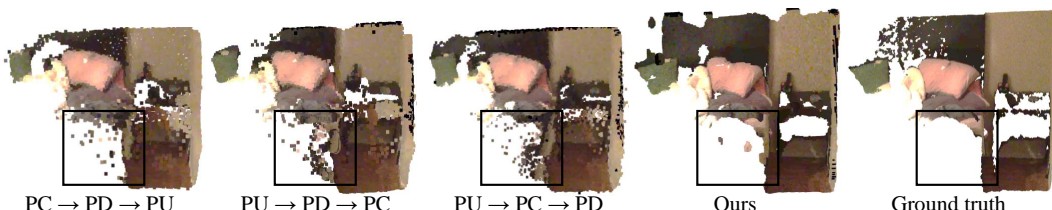

Figure 6: **Point cloud reconstruction results using three baselines.** Note that point cloud has been colorized for a visualization purpose.

| Type of positional encoding | | | | Performance | | preserve (✓) | | | Performance | |
|---|---|---|---|---|---|---|---|---|---|---|
| No PE | $\gamma_{\text{PE}}$ | PE* | $\gamma_{\text{ampPE}}$ | Chamf. | $f$-score | 1$^{\text{st}}$ stage | Self-attn | Cross-attn | Chamf. | $f$-score |
| ✓ | | | | 1.231 | 52.011 | ✓ | | | 1.42 | 49.56 |
| | ✓ | | | 1.224 | 52.434 | ✓ | ✓ | | 1.209 | 53.374 |
| | | ✓ | | 1.226 | 52.954 | ✓ | | ✓ | 1.223 | 52.692 |
| | | | ✓ | 1.195 | 55.488 | ✓ | ✓ | ✓ | 1.195 | 55.488 |
| (a) Ablation study for amplified positional encoding. | | | | | | (b) Ablation study for voxel re-localization. | | | | |

Table 2: **Ablation study of our point cloud reconstruction network.** In (a), PE* indicates positional encoding from NeRF Mildenhall et al. (2020). In (b), S-hourglass represents the voxel generation network. Self-attn and Cross-attn mean self-attention layers and cross-attention layers, respectively. The evaluation is conduced in ShapeNet-Part dataset under $l_{\text{vox}}$=0.02, $\mathcal{P}_{\text{in}}$=2048.

## 4.5 ABLATION STUDY

In this section, we propose to evaluate the contribution of each proposed component (amplified positional encoding and voxel re-localization network), through an extensive ablation study in Table 2.

In Table 2-(a), we validate our amplified positional encoding in comparison with different styles of positional encodings, $\gamma_{\text{PE}}$ (Vaswani et al., 2017) and PE* (Mildenhall et al., 2020). Also, we include an additional experiment that does not use any positional encodings (*i.e.*, No PE in Table 2-(a)). We measure the accuracy by calculating the Chamfer distance and $f$-score. It demonstrates that our method outperforms the previous embedding schemes. We deduce that by using the geometric position priors of point cloud, we can intentionally magnify or suppress the position encoding vectors that are usually beneficial for the control of high-frequency embedding in 3D spatial domain.

In Table 2-(b), we conduct experiments on different composition of voxel re-localization network. Our network involves sparse stacked hourglass network (*i.e.*, S-Hourglass), self-attention layers (*i.e.*, Self-attn in Table 1), and cross-attention layers (*i.e.*, Cross-attn in Table 1). By intentionally omitting certain modules in the voxel re-localization network, we measure the quality of point reconstruction using the Chamfer distance and $f$-scores. Despite reasonable results from the voxel generation network (*i.e.* 1$^{\text{st}}$ stage in Table 1), it turns out that self- and cross-attention layers further increase the accuracy of of the point cloud reconstruction. In particular, the combined usage of self- and cross-layers outperforms other strategies. To ensure a fair comparison, the self-attention only method and cross-attention solely network adopts more transformers to balance with our final network design.

So far, we demonstrate the effectiveness of our network design through a fair comparison with previous studies and an extensive ablation studies. Our network aims at point cloud reconstruction that can be applied in general cases to achieve high-quality point refinement.

## 5 CONCLUSION

Contrary to previous approaches – individually addressing the problems of densification and denoising, we propose an elegant two-stage pipeline to jointly resolves these tasks. As a result of this unified framework, the quality of sparse, noisy and irregular point cloud can be drastically improved. This claim has been validated through a large series of experiments underlying the relevance and efficiency of this joint refinement strategy. Specifically, qualitative and quantitative results demonstrate significant improvements in terms of reconstruction's accuracy and generalization when compared with prior techniques. Our point cloud reconstruction strategy paves the way towards more flexible, multi-tasks and effective architectures to refine point cloud's quality. Despite this success, there exist remaining issues, such as noise modeling for better denoising and point-voxel fusion design.

## ACKNOWLEDGMENTS

This work was supported by (1) NAVER LABS Corporation [SSIM: Semantic and scalable indoor mapping], (2) IITP grant funded by the Korea government(MSIT) (No.2019-0-01906, Artifcial Intelligence Graduate School Program(POSTECH)) and (3) Institute of Information and communications Technology Planning and Evaluation (IITP) grant funded by the Korea government(MSIT) (No.2021-0-02068, Artificial Intelligence Innovation Hub)

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

APPENDIX

In this appendix, we describe further details of the proposed methodology, deep point cloud reconstruction. First, we provide the additional quantitative results (Sec. A). Second, we describe precise equations of the metrics that we used for evaluation (Sec. B). Third, we present qualitative comparisons of ours and baseline methods (Sec. C).

## A QUANTITATIVE RESULTS

Along with Table 1 of the manuscript, we provide additional results that we did not include in the manuscript.

| Methods | $\mathcal{P}_{in}$ | $\mathcal{P}_{out}$ | k Nearest Chamf. ($\downarrow$) | | | Percentage ($\uparrow$) ($< 0.5$) | | | Percentage ($\uparrow$) ($< 1.0$) | | |
|---|---|---|---|---|---|---|---|---|---|---|---|
| | | | $k$=1 | $k$=2 | $k$=4 | Acc. | Comp. | $f$-score | Acc. | Comp. | $f$-score |
| **ShapeNet-Part dataset** | | | | | | | | | | | |
| PC ($r$=4) | 2048 | 8192 | 1.14 | 1.29 | 1.51 | 72.28 | 42.71 | 51.51 | 92.41 | 88.48 | 90.16 |
| PU ($r$=4) | 2048 | 8192 | 1.56 | 1.69 | 1.90 | 60.71 | 26.91 | 36.04 | 86.67 | 68.19 | 75.60 |
| Ours ($l_{vox}$=0.0200) | 2048 | 8732 | 1.19 | 1.33 | 1.52 | 81.02 | 40.41 | 53.48 | 96.95 | 81.23 | 88.08 |
| PC ($r$=8) | 2048 | 16384 | 1.01 | 1.13 | 1.31 | 70.89 | 60.66 | 64.17 | 92.16 | 93.56 | 92.76 |
| PC ($r$=8) $\rightarrow$ PD | 2048 | 16384 | 1.25 | 1.34 | 1.49 | 61.11 | 42.98 | 49.74 | 91.73 | 81.01 | 85.83 |
| PC ($r$=4) $\rightarrow$ PU ($r$=4) $\rightarrow$ PD | 2048 | | 1.14 | 1.48 | 1.60 | 56.55 | 49.72 | 52.43 | 87.13 | 82.73 | 84.67 |
| PU ($r$=8) | 2048 | 16384 | 1.02 | 1.14 | 1.32 | 80.84 | 50.36 | 60.74 | 95.69 | 82.21 | 88.05 |
| PU ($r$=8) $\rightarrow$ PD | 2048 | 16384 | 1.29 | 1.39 | 1.56 | 71.79 | 43.88 | 53.57 | 94.68 | 75.41 | 83.55 |
| PU ($r$=4) $\rightarrow$ PC ($r$=4) $\rightarrow$ PD | 2048 | | 1.51 | 1.59 | 1.71 | 56.63 | 59.90 | 57.63 | 83.13 | 89.47 | 85.97 |
| Ours ($l_{vox}$=0.0150) | 2048 | 15863 | 0.90 | 0.94 | 1.12 | 85.77 | 69.01 | 78.14 | 97.11 | 94.98 | 96.16 |
| **ScanNet dataset** | | | | | | | | | | | |
| PC ($r$=8) | 4098 | 32768 | 3.64 | 3.98 | 4.43 | 19.50 | 8.06 | 10.53 | 45.32 | 40.23 | 41.19 |
| PC ($r$=8) $\rightarrow$ PD | 4098 | 32768 | 3.61 | 3.84 | 4.19 | 24.58 | 10.33 | 13.82 | 55.21 | 38.18 | 44.47 |
| PU ($r$=8) | 4096 | 32768 | 3.20 | 3.49 | 3.92 | 21.52 | 9.323 | 11.94 | 48.30 | 36.10 | 39.72 |
| PU ($r$=8) $\rightarrow$ PD | 4098 | 32768 | 3.01 | 3.29 | 3.71 | 30.24 | 14.07 | 18.04 | 63.36 | 42.15 | 49.68 |
| Ours ($l_{vox}$=0.0100) | 4096 | 27740 | 1.51 | 1.64 | 1.87 | 70.49 | 40.59 | 51.29 | 93.63 | 73.24 | 82.02 |
| Ours ($l_{vox}$=0.0085) | 4096 | 41022 | 1.73 | 1.86 | 2.05 | 63.67 | 38.55 | 47.77 | 91.03 | 70.67 | 79.36 |
| PC ($r$=16) | 4098 | 65536 | 3.79 | 4.08 | 4.48 | 16.91 | 11.96 | 13.12 | 40.45 | 53.70 | 45.45 |
| PC ($r$=16) $\rightarrow$ PD | 4098 | 65536 | 3.71 | 3.93 | 4.23 | 21.93 | 15.05 | 17.33 | 51.57 | 47.38 | 49.12 |
| PC ($r$=4) $\rightarrow$ PU ($r$=4) $\rightarrow$ PD | 4096 | | 3.69 | 3.91 | 4.23 | 23.05 | 12.59 | 15.31 | 49.75 | 42.24 | 44.84 |
| PU ($r$=16) | 4096 | 65536 | 3.68 | 3.94 | 4.32 | 16.81 | 13.48 | 13.90 | 37.89 | 44.60 | 40.04 |
| PU ($r$=16) $\rightarrow$ PD | 4098 | 65536 | 2.86 | 3.10 | 3.44 | 26.06 | 21.04 | 22.36 | 55.73 | 53.56 | 54.04 |
| PU ($r$=4) $\rightarrow$ PC ($r$=4) $\rightarrow$ PD | 4096 | | 2.66 | 2.86 | 3.24 | 22.89 | 23.30 | 21.86 | 49.64 | 64.11 | 55.40 |
| PU ($r$=4) $\rightarrow$ PD $\rightarrow$ PC ($r$=4) | 4096 | | 3.75 | 3.99 | 4.33 | 15.99 | 15.43 | 14.50 | 36.60 | 54.70 | 43.10 |
| Ours ($l_{vox}$=0.0075) | 4096 | 55188 | 1.91 | 2.04 | 2.24 | 57.88 | 36.83 | 44.72 | 88.06 | 68.34 | 76.72 |
| Ours ($l_{vox}$=0.0065) | 4096 | 76182 | 2.17 | 2.32 | 2.32 | 51.55 | 34.78 | 41.16 | 84.47 | 65.28 | 73.37 |
| Ours ($l_{vox}$=0.0050) | 4096 | 116266 | 2.86 | 3.02 | 3.26 | 40.42 | 30.24 | 34.08 | 76.37 | 58.12 | 65.61 |
| **ICL-NUIM dataset** | | | | | | | | | | | |
| PC ($r$=8) | 4098 | 32768 | 3.32 | 3.59 | 3.97 | 21.61 | 12.28 | 13.87 | 44.32 | 50.18 | 45.98 |
| PC ($r$=8) $\rightarrow$ PD | 4098 | 32768 | 3.21 | 3.40 | 3.69 | 28.84 | 15.54 | 18.89 | 56.27 | 48.97 | 51.88 |
| PU ($r$=8) | 4096 | 32768 | 3.25 | 3.49 | 3.86 | 21.00 | 13.05 | 14.16 | 43.31 | 42.44 | 41.46 |
| PU ($r$=8) $\rightarrow$ PD | 4098 | 32768 | 2.85 | 3.09 | 3.45 | 32.34 | 20.03 | 22.75 | 61.38 | 51.12 | 54.96 |
| Ours ($l_{vox}$=0.0100) | 4096 | 21395 | 1.50 | 1.61 | 1.79 | 75.81 | 48.13 | 58.44 | 92.99 | 74.57 | 82.40 |
| Ours ($l_{vox}$=0.0085) | 4096 | 30986 | 1.70 | 1.81 | 2.00 | 71.65 | 46.43 | 55.74 | 91.01 | 72.36 | 80.12 |
| PC ($r$=16) | 4098 | 65536 | 3.41 | 3.63 | 3.95 | 18.69 | 17.53 | 16.65 | 38.26 | 63.02 | 47.20 |
| PC ($r$=16) $\rightarrow$ PD | 4098 | 65536 | 3.18 | 3.34 | 3.59 | 24.46 | 20.88 | 21.53 | 50.98 | 57.43 | 53.84 |
| PC ($r$=4) $\rightarrow$ PU ($r$=4) $\rightarrow$ PD | 4096 | | 3.61 | 3.81 | 4.42 | 21.62 | 11.41 | 14.05 | 45.34 | 42.57 | 43.32 |
| PU ($r$=16) | 4096 | 65536 | 3.83 | 4.03 | 4.33 | 17.78 | 18.19 | 16.83 | 35.45 | 49.93 | 41.03 |
| PU ($r$=16) $\rightarrow$ PD | 4098 | 65536 | 2.80 | 3.00 | 3.29 | 27.46 | 27.11 | 26.18 | 53.15 | 60.50 | 56.26 |
| PU ($r$=4) $\rightarrow$ PC ($r$=4) $\rightarrow$ PD | 4096 | | 3.61 | 3.94 | 4.42 | 27.55 | 8.64 | 12.42 | 54.42 | 33.58 | 40.60 |
| PU ($r$=4) $\rightarrow$ PD $\rightarrow$ PC ($r$=4) | 4096 | | 4.00 | 4.21 | 4.53 | 10.09 | 11.64 | 9.51 | 22.70 | 48.52 | 30.25 |
| Ours ($l_{vox}$=0.0075) | 4096 | 41322 | 1.87 | 1.99 | 2.18 | 67.78 | 44.97 | 53.36 | 89.09 | 70.52 | 78.141 |
| Ours ($l_{vox}$=0.0065) | 4096 | 56902 | 2.14 | 2.26 | 2.45 | 63.40 | 42.85 | 80.26 | 86.84 | 67.78 | 75.41 |
| Ours ($l_{vox}$=0.0050) | 4096 | 99321 | 2.78 | 2.92 | 3.10 | 54.11 | 38.09 | 43.55 | 81.76 | 61.88 | 69.45 |

Table 3: **Additional quantitative results.** We measure the quality of point reconstruction from ours, point completion (PC) (Xiang et al., 2021), point denoising (PD) (Luo & Hu, 2021) and point upsampling (PU) (Li et al., 2021) using typical criteria: chamfer distance (Fan et al., 2017) and percentage metric that consists of accuracy (Acc.) and completeness (Comp.) and $f$-score (Jensen et al., 2014). For fair comparison, we test various conditions by changing the number of input point cloud $\mathcal{P}_{in}$, the upsample ratio $r$, and voxel size $l_{vox}$. Note that identical colors mean the results from the same pre-trained weights for each method.

# B METRICS

## B.1 PERCENTAGE METRICS

Let us describe the precise equation of the metrics that we used in Table 1 of the manuscript and Table 3 of the appendix. Let $\mathcal{P}_{\text{GT}}$ be ground truth point cloud, $\mathcal{P}_{\text{pred}}$ a reconstructed points. For each inferred point $\mathbf{p} \in \mathcal{P}_{\text{pred}}$, its distance to ground truth is defined as:

$$d_{\mathbf{p} \to \mathcal{P}_{\text{GT}}} = \min_{\mathbf{g} \in \mathcal{P}_{\text{GT}}} \|\mathbf{p} - \mathbf{g}\|_2^2, \tag{6}$$

where $\mathbf{g}$ is a single point included in the ground truth point cloud $\mathcal{P}_{\text{GT}}$. These distances are gathered to calculate Accuracy (i.e., Acc.) as:

$$\text{Accuracy}(d_{\text{thresh}}) = \frac{100}{\mathcal{N}_{\mathcal{P}_{\text{pred}}}} \sum_{\mathbf{p} \in \mathcal{P}_{\text{pred}}} \left[ d_{\mathbf{p} \to \mathcal{P}_{\text{GT}}} \leq d_{\text{thresh}} \right], \tag{7}$$

where $\mathcal{N}_{\mathcal{P}_{\text{pred}}}$ is the number of reconstructed points $\mathcal{P}_{\text{pred}}$, $[\cdot]$ is the Iverson bracket, and $d_{\text{thresh}}$ is a threshold distance. In a similar manner, we calculate the distance from ground truth point $\mathbf{g} \in \mathcal{P}_{\text{GT}}$ to the reconstructed points $\mathcal{P}_{\text{pred}}$ as:

$$d_{\mathbf{g} \to \mathcal{P}_{\text{pred}}} = \min_{\mathbf{p} \in \mathcal{P}_{\text{pred}}} \|\mathbf{g} - \mathbf{p}\|_2^2, \tag{8}$$

These distances are gathered to calculate Completeness (i.e., Comp.) as:

$$\text{Completeness}(d_{\text{thresh}}) = \frac{100}{\mathcal{N}_{\mathcal{P}_{\text{GT}}}} \sum_{\mathbf{g} \in \mathcal{P}_{\text{GT}}} \left[ d_{\mathbf{g} \to \mathcal{P}_{\text{pred}}} \leq d_{\text{thresh}} \right] \tag{9}$$

where $\mathcal{N}_{\mathcal{P}_{\text{GT}}}$ is the number of ground truth point clouds. To calculate f1-score, we use Accuracy and Completeness as below:

$$\text{f1-score}(d_{\text{thresh}}) = \frac{2 \cdot \text{Accuracy}(d_{\text{thresh}}) \cdot \text{Completeness}(d_{\text{thresh}})}{\text{Accuracy}(d_{\text{thresh}}) + \text{Completeness}(d_{\text{thresh}})}, \tag{10}$$

Note that in ShapeNet dataset, we set $d_{\text{thresh}}{=}0.0200$. For ScanNet and ICL-NUIM dataset, we set $d_{\text{thresh}}{=}0.0050$ for evaluation. We mostly follow these metrics from (Jensen et al., 2014; Aanæs et al., 2016; Yao et al., 2018; Knapitsch et al., 2017).

## B.2 CHAMFER DISTANCE

Before we explain the $k$-**Nearest Chamfer distance**, let us revisit the original Chamfer distance. Based on the equations Eq. 8 and Eq. 6, we describe the Chamfer distance (i.e., Chamf.) as:

$$\text{Chamf.} = \frac{1}{\mathcal{N}_{\mathcal{P}_{\text{pred}}}} \sum_{\mathbf{p} \in \mathcal{P}_{\text{pred}}} (d_{\mathbf{p} \to \mathcal{P}_{\text{GT}}}) + \frac{1}{\mathcal{N}_{\mathcal{P}_{\text{GT}}}} \sum_{\mathbf{g} \in \mathcal{P}_{\text{GT}}} (d_{\mathbf{g} \to \mathcal{P}_{\text{pred}}}), \tag{11}$$

Since the original Chamfer distance only considers the closest distances, it has difficulty in describing the optimal distance between two point sets having different number of points (Fan et al., 2017). However, rather than to use Earth Mover distance that requires quadratic computational power and memory consumption, we formulate $k$-Nearest Chamfer distance by considering the $k$ closest points instead of the single closest point (i.e., $k{=}1$). Let us define the distance from a inferred point $\mathbf{p} \in \mathcal{P}_{\text{pred}}$ to the $k$ closest ground truth points $\mathbf{g}_k \in \mathcal{P}_{\text{GT}}$ as follow:

$$d_{\mathbf{p} \to \mathcal{P}_{\text{GT}}}^k = \frac{1}{k} \sum_k \|\mathbf{p} - \mathbf{g}_k\|_2^2, \tag{12}$$

where $\mathbf{g}_k$ is the $k$-th closest point to the inferred point $\mathbf{p}$. Similarly, we calculate the distance from a ground truth point $\mathbf{g}$ to the $k$-closest reconstructed points $\mathbf{p}_k$ as follows:

$$d_{\mathbf{g} \to \mathcal{P}_{\text{pred}}}^k = \frac{1}{k} \sum_k \|\mathbf{g} - \mathbf{p}_k\|_2^2, \tag{13}$$

Based on these equations, we compute $k$-Nearest Chamfer distance (i.e., $k$-Chamf.) as below:

$$k\text{-Chamf.} = \frac{1}{\mathcal{N}_{\mathcal{P}_{\text{pred}}}} \sum_{\mathbf{p} \in \mathcal{P}_{\text{pred}}} (d_{\mathbf{p} \to \mathcal{P}_{\text{GT}}}^k) + \frac{1}{\mathcal{N}_{\mathcal{P}_{\text{GT}}}} \sum_{\mathbf{g} \in \mathcal{P}_{\text{GT}}} (d_{\mathbf{g} \to \mathcal{P}_{\text{pred}}}^k). \tag{14}$$

# C  QUALITATIVE RESULTS

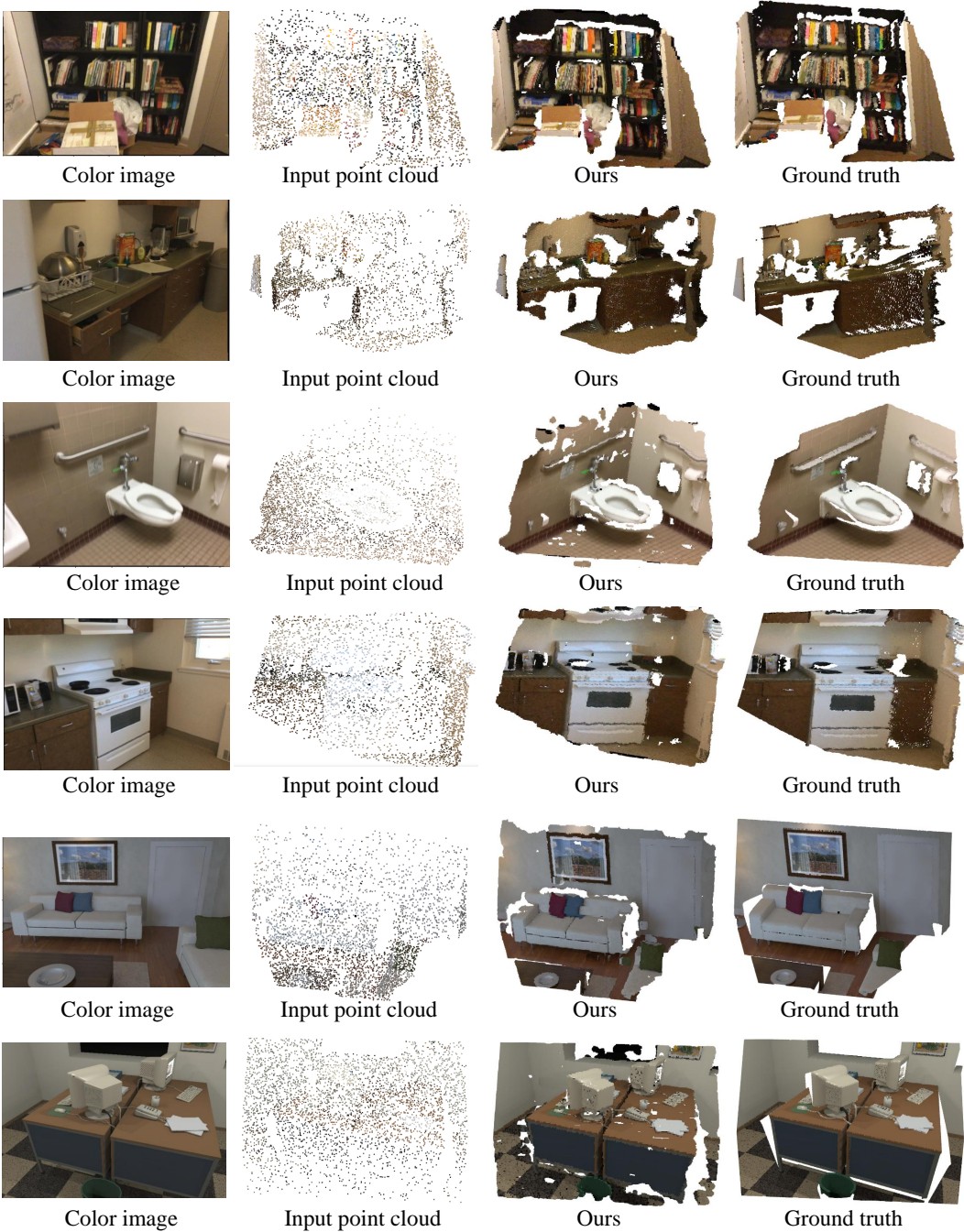

Figure 7: **Point cloud reconstruction results.** Note that point cloud has been colorized for a visualization purpose.

