# OpenReview forum: "Deep Point Cloud Reconstruction"
_ICLR.cc/2022/Conference — ICLR 2022 Poster_

### Official Review · Reviewer_dx4y · 2021-10-31

**Correctness:** 3
**Technical Novelty And Significance:** 2
**Empirical Novelty And Significance:** 2
**Recommendation:** 5
**Confidence:** 4

**Main Review:**

The problem of point cloud reconstruction combines several existing important challenges in point cloud processing. The paper is well written and easy to follow, however, some details remain unclear and the evaluation choices inconclusive.

Major issues:


- Since many scans are combined into a single scan, there is another challenge that is entirely overlooked - misalignment. If the idea is to combine all challenges into one, this should be another challange to address.
- Missing related work - There is a very large body of recent works on neural shape implicit representation which focuses on the surface reconstruction task. While I understand how this is a bit different from the proposed point cloud reconstruction, one could simply sample points on the zero level set and achieve point cloud reconstruction. Among these works are DeepSDF, Occupancy networks, SAL, SALD, DPDist, IGR,  Convolutional occupancy networks, PHASE, SIREN, DiGS, etc. These are missing particularly since in section 3.2 it is claimed that the local 3D shape is estimated using neighbour voxels.
- Approach - in the described approach it was not clear to me how the final number of points is specified. Also, not clear how color information is determined for the newly generated points
- Evaluation baselines: I find the choice of the baseline questionable. First, there was no unified approach used (denoising + completion + sampling). Second, It would make more sense to me to compare the proposed method on the other's task since it is expected that a denoising method will not do well for upsampling. Third, in the combined baselines (Snowflake-PC->Score-PD and Dis-PU->scorePD) it is not clear the order of application (it makes sense to first denoise and then upsample/complete but the text suggests the opposite which should also be reported). Finally, as mentioned in the related work comment, the paper lacks comparison to implicit representations surface reconstruction.
- Evaluation datasets - The chosen datasets are commonly used and an understandable choice for this task. However, they are strongly biased towards planar geometries. For example, most shapes in ShapeNet and most scenes in ScanNet have large planes. I expected to see results on the Surface Reconstruction Benchmark (Berger et. al.) which is small but has a mix of complex geometries.
-Evaluation - time and memory requirements are missing. From the implementation details it seems to be resource-heavy, how does that compare to the other baselines?
- Lack of insight - The paper is missing an important subsection that provides some insight into what the different modules have learned and how is that beneficial to the task. This raises some question about novelty - how is this approach better than the sum of its technical parts?
- In its current form, the paper is not self-contained and the method is not reproducible due to a lack of technical details.

Minor issues:
 - There is an issue with the term "continuous 3D points" which repeats itself in the paper (section 3.2 for example). Point clouds are not continuous, they are sampled on a continuous surface.
- Evaluation: The evaluation metrics were not properly specified. When performing standardized experiments it may be sometimes acceptable to commit this, however, in these case, the paper proposes a new task, therefore, the evaluation measures should be specified.
- $V_{mid}$ is missing from Fig 2 and Fig3. One can only assume where it is in the architecture. It should be better specified.
- Section 2 -> Point cloud upsampling "input points input points" typo.



**Summary Of The Paper:**

The paper proposes a new method to jointly address the tasks of point cloud denoising, completion and upsampling. They call this joint problem "Point cloud reconstruction". The method consists of two main modules - a voxel generation module (to increase voxel density and remove outliers) implemented as an hourglass sparse convolutional neural network and a point relocalization module (to convert the discrete voxels back to point clouds) implemented using a transformer network. The results show good performance in the chosen metrics and datasets.

**Summary Of The Review:**

In summary, I find the paper well written and the problem interesting. The approach is well presented and technically sound for the most part. However, due to the lack of a significant body of works in the related work section as well as the missing evaluations I can not recommend this paper for acceptance at this stage.

---

> ### Author Response · Authors · 2021-11-22
> **Response to Reviewer dx4y**
>
> **[Writing] Q.4-6.**
>  In its current form, the paper is not self-contained and the method is not reproducible due to a lack of technical details
>
> **A.4-6.**
> Following the request from the reviewer, we newly update Fig. 3 and Fig. 4 of the manuscript. Every detail of channel dimensions and connectivity between layers is precisely described. Moreover, we are planning to release the code after the review. **Since this is a pioneering paper that opens a new task, as a researcher, we think it is our responsibility to clearly open everything for the community.**
>
> ---
>
> **[Minor-Writing] Q.4-7.**
>  There is an issue with the term *continuous 3D points* which repeats itself in the paper. Point clouds are not continuous, they are sampled on a continuous surface.
>
> **A.4-7.**
> We apologize for the confusing terminology. We revise all these words in the manuscript.
>
>
> ---
>
> **[Minor-Metrics] Q.4-8.**
> The evaluation metrics were not properly specified. When performing standardized experiments it may be sometimes acceptable to commit this, however, in these cases, the paper proposes a new task, therefore the evaluation metrics should be specified.
>
> **A.4-8.**
> Following the request by the reviewer, we describe the precise equations of all metrics that we used in the appendix.
>
> ---
>
> **[Minor-Metrics] Q.4-9.**
> $\mathcal{V}_\text{mid}$ is missing.
>
> **[Minor-Metrics] A.4-9.**
> We apologize for the missing details. Following the reviewer's request, we newly update the Fig. 3 of the manuscript. Note that $\mathcal{V}_\text{mid}$ is the intermediate results from each hourglass network in the $1^\text{st}$ stage network.

---

> ### Author Response · Authors · 2021-11-22
> **Response to Reviewer dx4y**
>
> **[Novelty] Q.4-5.**
>  This paper is missing an important subsection that provides some insight into what the different modules have learned and what is that beneficial to the task. This raises some questions about novelty - how is this approach better than the sum of its technical parts?
>
> **A.4-5.**
> As noted by three reviewers (veKS, 1Mes, and G4Aq), our contributions derive from a two-stage pipeline that aims to handle raw point clouds via voxelization. This module is very simple and straightforward, but as we describe in Sec. 3.1 of the manuscript, each layer are deeply related to the point cloud reconstruction task. This architectural design leads to better generalization and transferability toward unmet and real-world 3D scans as in Table 4. Let us briefly explain the reasons behind our strength of generalization and transferability. Throughout this paper, we conjecture that this transferability can be explained by two dominant reasons.
>
> First, we optimize the nature of the objective function. Indeed improving the proximity-to-surface instead of task-specific losses (such as denoising, densification, and completion) seems to be a more generic and generalizable problem. Regardless of the difference in synthetic and real-world noises, our network is trained to improve the quality of reconstruction. Also, this is clearly different from the point completion task (Yuan et al., 2018) that concentrates on category-specific datasets for generating points in occluded regions of objects.
>
> Second, we utilize the hash table for sparse convolution layer instead of $k$ Nearest Neighbor. To handle noisy and sparse point cloud, $k$NN is not a proper choice since it is sensitive to the density of point clouds (Mao et al., 2019). Nonetheless, previous studies (Li et al., 2021; Xiang et al., 2021; Ye et al., 2018b) mainly rely on this clustering algorithm. In contrast, sparse convolution layer takes absolute scale of receptive field regardless of the input distribution, which results in better transferability toward unmet and real-world scenes.
>
> Nonetheless, we find that we need to further provide a reasonable analysis of our strength in generalization ability and transferability. Following the advice from the reviewer dx4y, we elaborate the strength of our method in Sec. 4.4 of the manuscript.

---

> ### Author Response · Authors · 2021-11-22
> **Response to Reviewer dx4y**
>
> **[Evaluation] Q.4-4.**
> I find the choices of the baseline questionable. First, there was no unified approach used (denoising + completion + sampling). Second, It would make more sense to me compare the proposed method on the other's task since it is expected that a denoising method will not do well for upsampling. Third, in the combined baselines (Snowflake-PC $\rightarrow$ Score-PD and Dis-PU $\rightarrow$ score-PD), it is not clear the order of application (it makes sense to first denoise and upsample/complete but the text suggest the opposite which should be reported). Finally, as mentioned in the related work, the paper lacks comparison to implicit representations surface reconstruction.
>
> **A.4-4.**
> Following the request of the reviewers, we conduct additional experiments about the combination of three models, denoising, completion and upsampling. It turns out that the different orders of the three models do not improve the quality of point reconstruction compared to the results in Table 1 of the manuscript. We claim that these results consistently support the necessity of a joint solution for point cloud reconstruction.
>
> We think that our point reconstruction technique cannot truly be considered as a multi-task learning pipeline since it solves the problems in a completely joined manner. As a result, we cannot isolate each individual task since our network does not have dedicated to design task-specific header networks nor task-oriented losses. Instead, we assume that these problems are inherent to point cloud obtained by scanning devices and should be resolved together. Moreover, even if we forcefully obtain the results in each single task, it will lack fairness and would not reflect our intention nor the capability of our point reconstruction technique. We hope to have clarified this particular point and that the reviewer will understand the context for which we have designed this point reconstruction strategy combining three tasks at once to resolve the inherent shortcomings of 3D point clouds.

---

> ### Author Response · Authors · 2021-11-22
> **Response to Reviewer dx4y**
>
> **[Details] Q.4-3.**
>  It was not clear to me how the final number of points is specified. Also, not clear how color information is determined for the newly generated points.
>
> **A.4-3.**
> As we described in Sec 3.1 of the manuscript, our network jointly generates and prunes voxels in an adaptive manner. Even though transposed generative convolution layer upsamples voxels in a fixed ratio of $r$, voxel pruning layer removes outliers based on their signals. Thus, we cannot strictly say the number of outputs and report the average of $\mathcal{P}_\text{out}$ in Table 1 of the manuscript.

---

> ### Author Response · Authors · 2021-11-22
> **Response to Reviewer dx4y**
>
> **[Related works] Q.4-2.**
> There is a very large body of recent works on neural shape implicit representation which focus on the surface reconstruction task.  ..., one could simply sample points on the zero level set and achieve point cloud reconstruction.
>
> **A.4-2.**
> We apologize for the missing references about surface reconstruction. Following the reviewer's advice, we included a paragraph in Sec. 2. However, we claim that the point cloud reconstruction task is clearly different from surface reconstruction. Let us briefly describe their difference.
>
> First, mesh triangulation is not related to the inherent shortcomings of point cloud. According to the recent papers, PointTriNet (Sharp & Ovsjanikov, 2020) and (Rakotosaona et al., 2021), conventional studies about point triangulation often require dense and uniform 3D points. It implies that the surface reconstruction task puts more weight on mesh generation from given 3D points.
>
> Second, according to the DeepSDF (Park et al., 2019), this paper also utilizes ShapeNet dataset (Chang et al., 2015), but relies on a large number of points for training, which is described as *To prepare data, we start by normalizing each mesh to a unit sphere and sampling **500,000** spatial points x’s: we sample more aggressively near the surface of the object as we want to capture a more detailed SDF near the surface.* Therefore, while being different, we believe that our research can be very complementary with such type of work. For instance, point cloud reconstruction can be applied as a pre-processing to provide a dense and well distributed point cloud to train DeepSDF.
>
> Third, point sampling from mesh cannot be ahead of point cloud reconstruction task. In our case, we utilize Poisson Disk Sampling (Yuksel, 2015; Zhou et al., 2108) to obtain ground truth point cloud from ground truth mesh provided in the ShapeNet dataset.
>
> Based on these three reasons, we claim that point cloud reconstruction has clear characteristics compared to the surface reconstruction task. Moreover, to obtain such a large amount of points to train recent implicit methods (Park et al., 2019), point cloud reconstruction should be taken as a pre-processing stage, ahead of surface reconstruction. We hope the reviewer is satisfied with the necessity of point cloud reconstruction task.

---

> ### Author Response · Authors · 2021-11-22
> **Response to Reviewer dx4y**
>
> **[Point cloud reconstruction] Q.4-1.**
> Since many scans are combined into a single scan, there is another challenge that is entirely overlooked, misalignment. If the idea is to combine all challenges into one, this should another challenge to address.
>
> **A.4-1.**
> Point cloud registration is another topic that is widely addressed in previous studies (Choy et al., 2020; Lu et al., 2019; Gojcic et al., 2020). However, in our opinion, point cloud registration (misalignment) is not an inherent problem of point clouds. Misalignment usually derives from the extrinsic parameters between multiple 3D scans. In our point of view, this problem happens when we fuse multiple 3D scans that are taken in different views or time stamps. We think that it will be a wonderful future direction that extends the applicability of point cloud reconstruction toward multiple 3D scans.

---

> ### Author Response · Authors · 2021-11-22
> **Response to Reviewer dx4y**
>
> Thank you for the precious comments. Following the reviewer's advice, we revise our initial manuscript as follows:
> - Elaborate the related works section about a surface reconstruction task.
> - Clarify the Sec. 3.2 of the manuscript.
> - Conduct experiments about possible combinations of three previous papers, which is added in Table 1 of the manuscript and Table 3 in the appendix.
> - Include description in Sec. 4.4 about the strength of the proposed method, generalization and transferability.
> - Add more qualitative results in the appendix.
> - Describe the precise equations of metrics in the appendix.
> - **Implementation will be released after the review.**
>
> With this revision, the initial manuscript becomes more reasonable, analytic and feasible. We hope that all reviewers are satisfied with the improved quality of the paper. **For the reviewers' convenience, we include our response letter in the appendix.**

---

### Official Review · Reviewer_G4Aq · 2021-11-02

**Correctness:** 4
**Technical Novelty And Significance:** 3
**Empirical Novelty And Significance:** 3
**Recommendation:** 8
**Confidence:** 4

**Main Review:**

This paper mainly has two strengths:
First, a two-stage pipeline is innovatively proposed, which can enable the network to solve multiple tasks such as point cloud denoising, completion and reconstruction at one time, and the reconstruction accuracy and visualization results of this method can surpass the latest methods in various sub fields.
Second, the innovative use of amplified positive encoding and transformers to compute the relationship between a center voxel and its neighbor voxels has certain enlightening significance for the follow-up work.

But at the same time, this paper also has some shortcomings.
From the experimental results, ShapeNet-Part dataset is comparable to SOTA results, and the generalization ability of ScanNet and ICL-NUIM datasets is significantly better than other methods, but the paper only compares the experimental results. It would be better if the paper could explain why the proposed new method can achieve such strong generalization ability, and why the generalization ability of the existing methods is poor.


**Summary Of The Paper:**

This paper proposes an elegant two-stage pipeline to jointly solve the three tasks of point cloud densification, denoising and reconstruction at one time. Voxel densification and denoising are performed using a 3D spark stacked hourglass network, and then voxels are reconstructed into dense point clouds using transformers. The experimental results of ShapeNet exceed most existing methods, and have strong generalization ability.

**Summary Of The Review:**

This paper creatively integrates different tasks such as point cloud denoising, completion and reconstruction into a two-stage task, and the generalization ability of the experimental results on ScanNet and ICL-NUIM datasets is much better than the latest existing methods. This paper creatively uses amplified positive encoding to compute the relationship between a center voxel and its neighbor voxels, which has certain enlightening significance for the follow-up work.

---

> ### Author Response · Authors · 2021-11-22
> **Response to Reviewer G4Aq**
>
> **[Generalization] Q.3-1.**
> It would be better if the paper could explain why the proposed new method can achieve such strong generalization ability, and why the generalization ability of the existing methods is poor.
>
> **A.3-1.**
> It is our pleasure for the reviewer to recognize the strength of our method, generalization.
> We conjecture that this transferability can be explained by two dominant reasons.
>
> First, we optimize the nature of the objective function. Indeed improving the proximity-to-surface instead of task-specific losses (such as denoising, densification, and completion) seems to be a more generic and generalizable problem. Regardless of the difference in synthetic and real-world noises, our network is trained to improve the quality of reconstruction. Also, this is clearly different from the point completion task (Yuen et al., 2018) that concentrates on category-specific datasets for generating points in occluded regions of objects.
>
> Second, we utilize the hash table for sparse convolution layer instead of $k$ Nearest Neighbor. To handle noisy and sparse point cloud, $k$NN is not a proper choice since it is sensitive to the density of point clouds (Mao et al., 2019). Nonetheless, previous studies (Li et al., 2021; Xiang et al., 2021; Ye et al., 2018b) mainly rely on this clustering algorithm. In contrast, sparse convolution layer takes absolute scale of receptive field regardless of the input distribution, which results in better transferability toward unmet and real-world scenes.
>
> These are our analyses of our strength in better generalization and transferability. With our sparse stacked hourglass network, we can achieve well-generalized reconstruction performance. These voxelized results are further refined by the $2^\text{nd}$ stage network that converts discrete voxels into 3D points.
>
> Finally, we revised Sec. 4.4 of the manuscript to clarify the discussion about generalization and transferability of our deep point cloud reconstruction network.

---

> ### Author Response · Authors · 2021-11-22
> **Response to Reviewer G4Aq**
>
> Thank you for the precious comments. Following the reviewer's advice, we revise our initial manuscript as follows:
> - Elaborate the related works section about a surface reconstruction task.
> - Clarify the Sec. 3.2 of the manuscript.
> - Conduct experiments about possible combinations of three previous papers, which is added in Table 1 of the manuscript and Table 3 in the appendix.
> - Include description in Sec. 4.4 about the strength of the proposed method, generalization and transferability.
> - Add more qualitative results in the appendix.
> - Describe the precise equations of metrics in the appendix.
> - **Implementation will be released after the review.**
>
> With this revision, the initial manuscript becomes more reasonable, analytic and feasible. We hope that all reviewers are satisfied with the improved quality of the paper. **For the reviewers' convenience, we include our response letter in the appendix.**

---

### Official Review · Reviewer_1Mes · 2021-11-03

**Correctness:** 3
**Technical Novelty And Significance:** 3
**Empirical Novelty And Significance:** 3
**Recommendation:** 6
**Confidence:** 4

**Main Review:**



Strengths:-
1. The paper brings a new perspective solving two old problems in computer vision, and this is useful. I like the idea it is clean and simple.
2. The two-stage novel architecture is simple, but apart from the additional positional embedding I don’t there is any other addition here. But the stage1 and stage2 architectures have been used in the past to solve problems. My understanding is that the network majorly befits from stage1 as a lot of qualitative results concentrate on completion.
3. The numbers are better and change with voxel resolution but significantly better than the baseline approaches.
4. Qualitative examples are good, and show the performance improvement but are limited, and would like to see more of them in the supplemental (which is missing)





Weakness:-
1. This paper needs a quite few presentation changes to make the idea clear and easy flowing. The initial section of the method section of the paper is up until 3.1 is clear and well explained while 3.2 needs a better explanation. I still do not understand the exact intuition behind the amplitude-based positional embedding and logic is not very clear.
2. The baselines are not explained well at all, I’m not sure if the comparison to the baselines is fair because the exact characteristics of the Snowflake-PC → Score-PD are not explained, and likewise for Dis-PU. These sections need rewriting.
3. There is no evaluation independently for Stage-1 all evaluations in the paper use Stage-1 and Stage-2. I think to justify the need for Stage-2 and see its impact it is important to set up some evaluation criteria for Stage-1 only. I understand that maybe doing the full evaluation on Stage-1 is not possible.
4. Reiterating a point I mentioned earlier → More qualitative examples are needed.
5. The paper claims like past work when they upsample they don’t have this scaling-up factor of $r$ but I think in this case this $r$ just manifests as the voxel density in stage 1.
6. I’m also not clear what happens when multiple voxel centers move to the same location in 3D points.






**Summary Of The Paper:**

The paper proposes an end-to-end unified approach to solve the subtasks in point cloud completion. The main selling point of the paper is that combine two existing methods for point cloud densification  & denoising (Stage 1) and point cloud completion (Stage 2). The paper does extensive experiments on 3 datasets and compares to 3 different baselines. They achieve SOTA performance on these datasets. The use of sparse convolution to process point clouds is quite interesting as an autoencoder (termed as hour glass) aligns with the task at hand.

**Summary Of The Review:**



Justification:-
I think the paper does interesting contributions but lacks extensive qualitative results and the “Snowflake-PC → Score-PD” and similar things make me wonder if things are fair. The performance gap between the method and baselines is quite huge and I’m not able to completely understand the exact difference in impact between Stage 1 and Stage 2. Depending upon the rebuttal by the authors I will change my rating upwards if given reasonable explanations.

---

> ### Author Response · Authors · 2021-11-22
> **Response to Reviewer 1Mes**
>
> **[Future direction] Q.2-6.**
> I am also not clear what happens when multiple voxel centers move to the same location in 3D points.
>
> **A.2-6.**
> We are pleased to answer this very well spotted observation. This is, indeed, a limitation of our architectural design. Specifically, when multiple points are closer than the voxel size, they are merged into a single voxel during the voxelization process. Although this was not clearly visible in our experiments, we agree that the raised concerns can harm the results. In the future, we think of a new network design that utilizes both voxel and point-level understanding to overcome such limitations.

---

> > ### Comment · Reviewer_1Mes · 2021-11-29
> > **Discussion of Limitations**
> >
> > Thank you for your response. I think it would be necessary to discuss this limitation in the paper for future works to address.

---

> ### Author Response · Authors · 2021-11-22
> **Response to Reviewer 1Mes**
>
> **[Details] Q.2-5.** The paper claims like past work when they upsample they do not have this scaling-up factor of $r$ but I think in this case, $r$ just manifests as the voxel density in stage 1.
>
> **A.2-5.**
> It is true that the *generative sparse transposed convolution layer* densifies voxels in a fixed ratio~$r$. Following the reviewer's comment, we revise the sentences in Sec. 3.1 of the manuscript.

---

> ### Author Response · Authors · 2021-11-22
> **Response to Reviewer 1Mes**
>
> **[Details] Q.2-4.** There is no evaluation independently for Stage-1 all evaluations. I understand that maybe doing the full evaluation on Stage-1 is not possible.
>
> **A.2-4.**
> The evaluation for our $1^\text{st}$ network is already included in Table2-(b) of the manuscript. For clarity, we denote the name as $1^\text{st}$ stage.

---

> ### Author Response · Authors · 2021-11-22
> **Response to Reviewer 1Mes**
>
> **[Writing] Q.2-3.**
>  The baselines are not explained well, the exact characteristics of the Snowflake-PC $\rightarrow$ Score-PD are not explained, likewise for Dis-PU. These sections need rewriting.
>
> **A.2-3.**
> As requested by the reviewer, we revised Sec. 4.2 and Sec. 2 of the manuscript. Also, we included additional experiments about ensembles of three related papers in Table 1 of the manuscript and Table 3 in the appendix. Let us present the revised paragraphs in Sec. 4.2 below.
>
> In particular, we analyze the efficacy of ours compared to the combination of the existing studies. For instance, in Table 1 of the manuscript, *PC($r{=}4$)${\rightarrow}$PU($r{=}4$)${\rightarrow}$PD* represents consecutive operation of point completion, and point upsampling and point denoising under upsampling ratio $r{=}4$. Note that we also take into account the point completion task since this task also considers point densification on visible surface.

---

> > ### Comment · Reviewer_1Mes · 2021-11-29
> > **Additional Evaluation**
> >
> > Thanks for setting this up. Could you also discuss how was $r=4$ chosen for PC and PU?

---

> ### Author Response · Authors · 2021-11-22
> **Response to Reviewer 1Mes**
>
> **[Amplified positional encoding] Q.2-2.**
> The initial section of the paper until sec. 3.1 is clear and well explained, while section 3.2 needs a better explanation. I still do not understand the exact intuition behind the amplitude-based positional embedding.
>
> **A.2-2.**
> We newly revise Sec. 3.2 of the manuscript following the reviewer's comment. The intuition behind our amplitude positional encoding is clear, **when to embed high frequency signal?** We are inspired by NeRF (Mildenhall et al., 2020) where positional encoding results in preserving the high-frequency image signal. In our case, we are in a different situation. high-frequency signal is not always welcomed. When a voxel needs to move to a distant location, it is better to keep the low frequency signal, which is simply conducted by decreasing the amplitude of the positional encoding vector as in Fig. 4 of the manuscript.
>
> Such control is possible since we already know the relative distance between voxels that are estimated from the $1^\text{st}$ stage network. As underlined by the reviewer G4Aq, we believe that our amplified encoding is an important contribution of this work. To better highlight the necessity of this component, we further elaborate the paragraphs about our amplified positional encoding in Sec 3.2 of the manuscript.

---

> ### Author Response · Authors · 2021-11-22
> **Response to Reviewer 1Mes**
>
> **[More qualitative results] Q.2-1.**
> Qualitative examples are good and would like to see more of them in the supplemental material.
>
> **A.2-1.**
> We newly include qualitative results in the appendix. Also, we update the results from sequential uses of three baselines in Fig. 5 of the manuscript.

---

> ### Author Response · Authors · 2021-11-22
> **Response to Reviewer 1Mes**
>
> Thank you for the precious comments. Following the reviewer's advice, we revise our initial manuscript as follows:
> - Elaborate the related works section about a surface reconstruction task.
> - Clarify the Sec. 3.2 of the manuscript.
> - Conduct experiments about possible combinations of three previous papers, which is added in Table 1 of the manuscript and Table 3 in the appendix.
> - Include description in Sec. 4.4 about the strength of the proposed method, generalization and transferability.
> - Add more qualitative results in the appendix.
> - Describe the precise equations of metrics in the appendix.
> - **Implementation will be released after the review.**
>
> With this revision, the initial manuscript becomes more reasonable, analytic and feasible. We hope that all reviewers are satisfied with the improved quality of the paper. **For the reviewers' convenience, we include our response letter in the appendix.**

---

### Official Review · Reviewer_veKS · 2021-11-04

**Correctness:** 3
**Technical Novelty And Significance:** 2
**Empirical Novelty And Significance:** 3
**Recommendation:** 6
**Confidence:** 3

**Details Of Ethics Concerns:**

Disclaimer: for research integrity issues (e.g., plagiarism, dual submission), I didn't check very carefully.  For other issues, I don't find this paper violates any.

For the chairs: if you need me to check the integrity issues, please recommend some tools/software/website so that I can do it systematically. Happy to do it.


**Main Review:**

Note: I'll use the the word "multi-task" to refer to the problem setting in this work, in contrast to the single-task baselines. I understand that "multi-task" slightly abused here as one may argue that the the studied problem is a unified single task rather than three independent ones.

Strengths:
1. The high-level idea makes lots of sense to me. Using voxelization to denoise and complete irregular point cloud is reasonable and smart. Besides, we can sample as many points as we want from a voxel grid to get a dense surface. I'm not 100% sure whether this idea has been studied before as I'm not an expert for these two tasks. If possible, the authors could further elaborate on this contribution.
2. The empirical results are strong. The proposed framework greatly out-performs single task baselines.
3. Writing is mostly clear and proper ablation has been conducted.

Weakness:
1. The presented results are great, but the baselines are not very considerate. This paper is arguing that the three tasks should be unified and hence multi-task joint-learning baselines should be considered. The simplest setting is to ensemble all three baselines together in an optimized order. Alternatives contain training each baseline methods on all three tasks jointly.
2. Following above point, the proposed method should also be evaluated for each single tasks to demonstrate advantages. For example,  I would be only curious about how good is the proposed framework for point completion if my point cloud isn't noisy. Does this method improved over specified point-completion network?
3. I'm not fully convinced by the story of positional encoding. In the original form, the distance information is already expressed as the frequency of the signal. The magnitude is just redundant information. Why is it important? And what kind of effect it will cause precisely?
4. Lots of details are unclear. I list several that confuses me most:
    - How many voxels are processed in the transformer on average? Since the complexity of the attention function is $O(N^2)$, does the proposed framework suffer from big memory consumption?
    - For the tested dataset, especially the synthetic ones, how does the input noise/sparsity/incompleteness are created? ScanNet has lots of incomplete objects (eg., chair with no legs) but I don't find any qualitative results showing that the network can complete them, which is unsatisfactory. Also, regular synthetic noise (eg.,  Gaussian noise) are much simpler than real-world noise. How does the proposed framework deal with real-world noise? For example, MVS reconstructed point-clouds. Similar thing also holds for sparsity. It's unclear how does this framework work for real-world sparse point clouds (eg., the remote areas in Lidar data). If the authors think these are out-of-the-scope, then please clearly introduce the problem setting and limitations.
   - How is the dimension of $P_out$ decided? I thought arbitrary number of points can be generated. If this is true, how are the numbers in Tab.1 decided? I found these values of $P_out$ are confusing and they are not consistent with prior work.

Misc
1. The difference between sparse (point up-sampling) and incomplete (point-completion) is not very clear. Very specific definition should be made in the very early stage of the writing so that there is no ambiguity between the challenges that these two problems pose. Besides, the experimental results should clearly demonstrate how does the proposed framework address each sub-problem (above two+de-noising) individually. Unfortunately, these results are either missing or not very clear right now.
2. If find it mis-leading to term this task "point-cloud reconstruction". A maybe improper analogy is that we don't call joint image de-noising and super-resolution and image generation. If the authors really want to use the term reconstruction, maybe call it "conditioned point cloud reconstruction"? Or something else that is more accurate like "point cloud refinement"? I'm just making examples and throwing out random names.
3. It's unclear to me what's the motivation of clustering (voxel hashing), and how important is this step? Why do we need a clustering before transformer?  And where are the cluster information used in the later framework? If I understand correctly, this is used because the voxel-to-point step only operates on center voxels and hence we want a locally augmented feature representation?

Minor issues:
1. Fig.2 input figure is barely visible. A better color-map should be used.
2. 3D sparse Hourglass network is not a super popular network. It's also not implemented in popular sparse conv net framework (eg., Minkowski engine). Why this framework is adopted? Maybe the authors can share some related work to justify the advantage. It's not a common choice.

Suggestions:
1. In Tab.1, maybe the light color can be used to high-light the 2nd best among the baseline method? High-lighting the 2nd best among all including "ours" cannot fully demonstrate the advantages.

**Summary Of The Paper:**

This paper presents a unified framework for doing point cloud upsampling, denoising, and completion jointly. The proposed framework contains two modules: a point-to-voxel auto-encoder, and a Point Re-localization transformer. It outperforms single-task baselines on multiple benchmarks by a great margin.

**Summary Of The Review:**

Summary: My current rating is mainly based on two reasons: 1) The experimental results are strong and the idea of utilizing voxelization for de-noising/refining/densify-ing point clouds makes lots of sense to me. In addition, the proposed transformer works well for further improving the results and recovering points from voxels. 2) Even though the numbers are great, I do think the experiments can be improved. Multi-task baselines should be devised and compared. It's not enough to only demonstrate the proposed method out-performs single-task baselines. Moreover, fine-grained experiments should also be conducted to help us understand the performance on each single tasks.

I'll also use this section to justify my ratings below. This paper is mainly proposing a new problem setting and develops a new framework correspondingly.  Technical wise, this new framework doesn't carry significant new components. It does a great job choosing the proper sub-modules,  combining and slightly improving them for the final tasks. However, most of these components existed in the literature. Empirical wise, the results in Tab.1 is exciting and the proposed framework greatly improves over singe-task baselines. More strictly speaking, single tasks and two-task (X--> score-PD) baselines. The qualitative results in Fig.5 are also encouraging and clearly show the advantages.

I have worked in the field of point-cloud recognition and pretty familiar with the recognition frameworks.  However, I'm not an expert of point-cloud de-noising/completion/densify-ing, and not very familiar with the SOTA methods in these fields.

---

> ### Author Response · Authors · 2021-11-22
> **Response to Reviewer veKS**
>
> ---
> **[highlight] Q.1-13.**
> In Tab.1, maybe the light color can be used to highlight the 2nd best among the baseline method. Highlighting the 2nd best among all including **ours** cannot fully demonstrate the advantages.
>
> **A.1-13.**
> Following the request by the reviewer, we remove the 2nd best highlight in Table 1 of the manuscript.

---

> ### Author Response · Authors · 2021-11-22
> **Response to Reviewer veKS**
>
> ---
> **[Figure] Q.1-12.**
> Fig2 input figure is barely visible.
>
> **A.1-12.**
> We are sincerely thankful regarding this comment on the editorial quality of the paper.  In Fig 2 of the manuscript, to faithfully highlight the ability of our network, we intentionally set the same point's size for the visualization of the input and output point clouds. This clearly emphasizes how precisely and densely our network reconstructs the given sparse points. To clarify this point, we include more details about visualization in Fig 2 of the manuscript.

---

> ### Author Response · Authors · 2021-11-22
> **Response to Reviewer veKS**
>
> **[Missing details] Q.1-11.**
> Where are the cluster information used in the later framework? If I understand correctly, this used because the voxel-to-point step only operates on center voxels and hence we want a locally augmented feature representation?
>
> **A.1-11**
> To be more informative, we revise the paragraphs in Sec.3.2.
>
> Let us describe the detailed process in the point re-localization network, which is illustrated in Fig. 4. Given output voxels, we collect the K(=8) closest voxels to each voxel using the hash table that we used in the 1st stage network. Then, we obtain a voxel set that is referenced to the query voxel. As in Fig. 4, the 2nd stage network regresses the location of a query point (red voxel) using the geometric relation between the query voxel and its neighbor (white voxels) through self/cross attention layers.
>
> While this formulation is close to PointTransformer (Zhao et al., 2021), our point re-localization network has three distinctive differences. First, PointTransformer utilizes *k*-Nearest Neighbors to group 3D points, we re-use the voxel hashing that is computed in the previous stage. Second, our network consists of both self-attention and cross-attention. In our analysis, it is better to use two different attention layers than to solely adopt self-attention.

---

> ### Author Response · Authors · 2021-11-22
> **Response to Reviewer veKS**
>
> **[Title] Q.1-10.**
> If find it misleading to their this task *point cloud reconstruction*. A maybe improper analogy is that we do not call joint image denoising and super-resolution and image generation. If the authors really want to use the term reconstruction, maybe call it *conditioned point cloud reconstruction*? Or something else that is more accurate like *point cloud refinement*?
>
> **A.1-10**
> We are pleased for the reviewer's recommendation. We also think of this naming issue before the submission. Our conclusion is that we put more weights on the quality of 3D reconstruction using 3D points, so called point cloud reconstruction. As addressed by the reviewer dx4y, this task can be similar to surface reconstruction task because of the common goals, high-quality 3D description. However, without point triangulation or mesh formulation, this task truely focuses on the point itself to describe 3D environments. We hope that the reviewer is satisfied with our opinion.

---

> ### Author Response · Authors · 2021-11-22
> **Response to Reviewer veKS**
>
> **[Related works] Q.1-9**
> The difference between point upsampling and point completion is not very clear. The very specific definition should be made in the very early stage of the writing so that there is no ambiguity between the challenges that these two problems pose.
>
> **A.1-9**
> The difference between the different tasks has been described in the related works section. The goal of point upsampling is to densify sparse but homogeneously distributed input points. On the other hand, the point completion task, as described in the pioneering paper (Yuan et al., 2018), aims at extrapolating entire parts of the objects resulting from occlusions or the limited sensor resolution. Regarding the experiments validating each task, we invite the reviewer to refer to our response entitled **Multi-task learning** in this rebuttal.

---

> ### Author Response · Authors · 2021-11-22
> **Response to Reviewer veKS**
>
> **[Missing details] Q.1-8.**
> How is the dimension of $\mathcal{P}_\text{out}$ decided?
>
> **A.1-8.**
> Since our 1st stage network adaptively generates and prunes voxels, we cannot precisely decide for the dimension of $\mathcal{P}_\text{out} {\in} \mathbb{R}^{N'{\times}3}$ where $N'$ is determined after the inference. Thus, we report the average cardinality of the generated point clouds in Table 3 of the manuscript.

---

> ### Author Response · Authors · 2021-11-22
> **Response to Reviewer veKS**
>
> **[Missing details] Q.1-7.**
> How many voxels are processed in the transformer on average?
>
> **A.1-7.**
> We apologize for the missing details. We set $k=8$ as an input of our transformer. This information is included in Sec. 3.2 of the manuscript and Table 3 of the appendix.

---

> ### Author Response · Authors · 2021-11-22
> **Response to Reviewer veKS**
>
> **[Sparse stacked hourglass network] Q.1-6.**
> 3D sparse stacked hourglass network is not a super popular network. It is also not implemented in the popular sparse conv net framework (e.g., Minkowski engine). Why this network is adopted? Maybe the authors can share some related works to justify the advantage. It is not a common choice.
>
> **A.1-6.**
> In our understanding, stacked hourglass has unique properties and is widely used for various applications. First of all, we found that the total number of citations of stacked hourglass network is **3364** (Newell et al., 2016b) and **541** (Newell et al., 2106a). Note that the original stacked hourglass network is proposed in (Newell et al., 2016b) and the modified version of the network is introduced in (Newell et al., 2106a) by the same first author.
>
> Moreover, the variants of stacked hourglass network largely used in various tasks. For instance in stereo matching network, PSMNet (cite:**706**) (Chang & Chen, 2018), GANet (cite:**246**) (Zhang et al., 2019), AANet (cite:**76**) (Xu & Zhang, 2020), utilizes a volumetric version of hourglass network using 3D CNNs. This paper clearly mentions their stacked hourglass network in their abstract, *"The 3D CNN learns to regularize cost volume using stacked multiple hourglass networks in conjunction with intermediate supervision."* In another example, multi-view stereo task, DPSNet (cite:**98**) (Im et al., 2019)  also adopts 3D stacked hourglass network to aggregate cost volumes. Different from deep and wide U-shape network, such as Minkowski-UNet (Choy et al., 2019), using multiple light encoder-decoder networks (i.e., stacked hourglass network) aims to sequentially refine feature vectors by intermediate losses.
>
> In summary, our $1^\text{st}$ stage network aims to sequentially refine the given point cloud through the proposed sparse stacked hourglass network. Similar to the idea of voxel carving, hourglass networks re-shape the sparse points in a coarse-to-fine manner. We hope that the reviewer veKS is satisfied with our understanding of stacked hourglass network (Newell et al., 2016b). Also, we expect the future studies about point cloud reconstruction will widely use this sparse version of stacked hourglass network as stereo matching methods and multi-view stereo studies do.

---

> ### Author Response · Authors · 2021-11-22
> **Response to Reviewer veKS**
>
> **[Generalization] Q.1-5.** For the tested dataset, especially this synthetic one (ICL-NUIM), how does the input noise/sparsity/incompleteness are created? Also, regular synthetic noise (e.g., Gaussian noise) are much simple than real-world noise. How does the proposed framework deal with real-world noise? It is unclear how does this framework work for real-world sparse point clouds (e.g., the remote areas in LiDAR data).
>
> **A.1-5.** Thank you very much for this remark regarding the transferability of the approach.
> As the reviewer veKS mentioned, we train our network only in synthetic/object-scale dataset (ShapeNetPart). Then, without fine-tuning, we utilize the pre-trained weights to measure the quality of reconstruction using unmet point clouds in ScanNet or ICL-NUIM datasets. Despite the domain gap between datasets, we think our network generalizes well toward various real datasets. We conjecture that this transferability can be explained by two dominant reasons.
>
> First, we optimize the nature of the objective function. Indeed improving the proximity-to-surface instead of task-specific losses (such as denoising, densification, and completion) seems to be a more generic and generalizable problem. Regardless of the difference in synthetic and real-world noises, our network is trained to improve the quality of reconstruction. Also, this is clearly different from the point completion task (Yuan et al., 2018) that concentrates on category-specific datasets for generating points in occluded regions of objects.
>
> Second, we utilize the hash table for sparse convolution layer instead of *k* Nearest Neighbor. To handle noisy and sparse point cloud, *k*NN is not a proper choice since it is sensitive to the density of point clouds (Mao et al., 2019). Nonetheless, previous studies (Li et al., 2021; Xiang et al., 2021; Ye et al., 2018b) mainly rely on this clustering algorithm. In contrast, sparse convolution layer takes absolute scale of receptive field regardless of the input distribution, which results in better transferability toward unmet and real-world scenes.
>
> Note that we apply random Gaussian noise into ICL-NUIM dataset following (Le et al., 2021). This synthetic noise is truely much simpler than the real-world noise, as addressed by the reviewer. Since we did not precisely analyze the noise modeling, we have improved the exposition in the paper to clarify the discussion.

---

> ### Author Response · Authors · 2021-11-22
> **Response to Reviewer veKS**
>
> **[Amplified positional encoding] Q.1-4.**
> I am not fully convinced by the story of positional encoding. In the original form, the distance information is already expressed as the frequency of the signal. The magnitude is just redundant information. Why is it important? And what kind of effect it will cause precisely?
>
> **A.1-4.**
> We apologize for the confusion. The intuition behind our amplitude positional encoding is the following, **when to embed high frequency signal?** We are inspired by NeRF (Mildenhall et al., 2020) where positional encoding results in preserving the high-frequency image signal. In our case, we are facing a different situation. high frequency signal is not always welcomed. When a voxel needs to move to a distant location, it is better to keep the low frequency signal, which is simply operated by decreasing the amplitude of the positional encoding vector as described in Fig. 4 of the manuscript.
>
> Such control is possible since we already know the relative distance between voxels that are estimated from the 1st stage network. As underlined by the reviewer G4Aq, we believe that our amplified encoding is an important contribution of this work. To better highlight the necessity of this component, we further elaborate the paragraphs about our amplified positional encoding in Sec. 3.2 of the manuscript.

---

> ### Author Response · Authors · 2021-11-22
> **Response to Reviewer veKS**
>
> **[Multi-task learning] Q.1-3.**
> The proposed method should also be evaluated for each single task to demonstrate advantages.
>
> **A.1-3.**
> We understand the intention of the reviewer veKS. We guess that the worrying points derive from the reviewer veKS's initial description, *I will use the word multi-task to refer to the problem setting in this work, in contrast to the single-task baselines.* If this paper can be viewed as multi-task learning of three different tasks, it is important to evaluate the results on each single task, such as denoise, densification and completion task.
>
> However, our point reconstruction technique cannot truly be considered as a multi-task learning pipeline. There are two reasons. First, the series of problems (sparsity, noise, and irregularity) are inherently resided in raw point clouds. In real-world point clouds, these problems are completely mixed. Second, we solve the problems in a completely joined manner. As a result, we cannot isolate each individual task since our network does not have dedicated to design task-specific header networks nor task-oriented losses. Moreover, even if we forcefully obtain the results in each single task, it will lack fairness and would not reflect our intention nor the capability of our point reconstruction technique. We hope to have clarified this particular point and that the reviewer will understand the context for which we have designed this point reconstruction strategy combining three tasks at once to resolve the inherent shortcomings of 3D point clouds.

---

> ### Author Response · Authors · 2021-11-22
> **Response to Reviewer veKS**
>
> **[Baseline] Q.1-2.**
> The presented results are great, but the baselines are not very considerate. This paper is arguing that the three tasks should be unified and hence multi-task joint-learning baselines should be considered. The simplest setting is to ensemble all three baselines together in an optimized order. Alternatives contain training each baseline method on all three tasks jointly.
>
> **A.1-2.**
> Admittedly, our initial experimental setup was lacking fairness, therefore, we have re-conducted a large set of experiments where all possible combinations between the three baselines have been investigated. The obtained results as visible in Table 1 of the manuscript and Table 3 of the appendix. Please take an example with some notations used in the tables in Table 1 of the manuscript.
> For instance, *PC(r=4)->PU(r=4)->PD* represents the consecutive operation of point completion, and point upsampling and point denoising under fixed upsampling ratio r=4. Note that we also take into account the point completion task since this task also considers point densification on the visible surfaces.
>
> None of the tested configurations is qualitatively nor quantitatively competitive with our point reconstruction technique. It should however be noted that since only 2048 points have been used as input, the point denoising method (Lue & Hu, 2021) could not be used before upsampling (because (Lue & Hu, 2021) requires at least 10,000 points to operate). Thus, we cannot provide results for every single possible combination. Nonetheless, our method still achieves state-of-the-art performance among the possible orders of point denoising/completion/upsample networks.

---

> ### Author Response · Authors · 2021-11-22
> **Response to Reviewer veKS**
>
> **[Writings] Q.1-1.**
> The high-level idea makes lots of sense to me. Using voxelization to denoise and a complete irregular point cloud is reasonable and smart. Besides, we can sample as many points as we want from a voxel grid to get a dense surface. If possible, the authors could further elaborate on these contributions.
>
> **A.1-1.**
> Thank you very much for this positive remark regarding using voxelization for this task. As suggested, we underlined the flexibility offered by our voxel-based technique in Sec. 4.4 of the manuscript. This advantage is also highlighted by the quantitative results presented in Table 1 of the manuscript and Table 3 of the appendix. These results demonstrate that the number of output points can be adjusted by the hyper-parameters while preserving the overall reconstruction quality.

---

> ### Author Response · Authors · 2021-11-22
> **Response to Reviewer veKS**
>
> Thank you for the precious comments. Following the reviewer's advice, we revise our initial manuscript as follows:
> - Elaborate the related works section about a surface reconstruction task.
> - Clarify the Sec. 3.2 of the manuscript.
> - Conduct experiments about possible combinations of three previous papers, which is added in Table 1 of the manuscript and Table 3 in the appendix.
> - Include description in Sec. 4.4 about the strength of the proposed method, generalization and transferability.
> - Add more qualitative results in the appendix.
> - Describe the precise equations of metrics in the appendix.
> - **Implementation will be released after the review.**
>
> With this revision, the initial manuscript becomes more reasonable, analytic and feasible. We hope that all reviewers are satisfied with the improved quality of the paper. **For the reviewers' convenience, we include our response letter in the appendix.**

---

### Author Response · Authors · 2021-11-22
**Response to AC and all Reviewers**

Thank you for the precious comments. Following the reviewer's advice, we revise our initial manuscript as follows:
- Elaborate the related works section about a surface reconstruction task.
- Clarify the Sec. 3.2 of the manuscript.
- Conduct experiments about possible combinations of three previous papers, which is added in Table 1 of the manuscript and Table 3 in the appendix.
- Include description in Sec. 4.4 about the strength of the proposed method, generalization and transferability.
- Add more qualitative results in the appendix.
- Describe the precise equations of metrics in the appendix.
- **Implementation will be released after the review.**

With this revision, the initial manuscript becomes more reasonable, analytic and feasible. We hope that all reviewers are satisfied with the improved quality of the paper. **For the reviewers' convenience, we include our response letter in the appendix.**

---

### Decision · Program_Chairs · 2022-01-20

**Decision:**

Accept (Poster)

**Comment:**

The paper proposes a unified framework for point cloud upsampling, denoising, and completion through a two-stage approach. It receives three reviews with three leaning to accept and one leaning to reject. Most of the reviewers like the proposed two-stage approach for its simplicity and demonstrated strong performance. The reviewer recommending marginally below the acceptance threshold expresses concerns about missing comparison to neural shape implicit representation and a lack of insights on what is learned by individual layers in the network. While the meta-reviewer agrees that having both would make the paper stronger, the meta-reviewer feels the paper has enough merit and would like to recommend its acceptance.